# Toward Long-Term Monitoring of Regional Permafrost Thaw with Satellite InSAR

Taha Sadeghi Chorsi[1], Franz J. Meyer[2], and Timothy H. Dixon[1]

[1]School of Geosciences, University of South Florida, Tampa, FL, USA
[2]Geophysical Institute, University of Alaska Fairbanks, AK, USA

**Correspondence:** Taha Sadeghi Chorsi (taha4@usf.edu)

**Abstract.** Active layer thickness (ALT) is estimated for a study area in Northern Alaska's continuous permafrost zone using satellite data from Sentinel-1 (radar) and ICESat-2 (LiDAR) for the period 2017 to 2022. Synthetic Aperture Radar (SAR) interferograms were generated using a Short Baseline Subset (SBAS) approach. Displacement time series over the thaw season (June-September) are well fit with a linear model (RMSE scatter is less than 7 mm) and show maximum seasonal subsidence of 20-60 mm. ICESat-2 products were used to validate the InSAR displacement time-series. ALT was estimated from measured subsidence using a widely used model exploiting the volume difference between ice and water, reaching a maximum depth in our study area of 1.5 m. Estimated ALT is in good agreement with in-situ and other remotely sensed data, but is sensitive to assumed thaw season onset, indicating the need for reliable surface temperature data. Our results suggest the feasibility of long-term permafrost monitoring with satellite InSAR. However, the C-band ($\sim$55 mm center wavelength) Sentinel radar is sensitive to vegetation cover, and in our studies was not successful for similar monitoring in the heavily treed discontinuous permafrost zone of Central Alaska.

## 1 Introduction

Permafrost is usually covered with soil or sediment – the active layer – which freezes and thaws seasonally. The annual freeze-thaw cycle causes surface height changes due to the volume difference between ice and liquid water. Active layer thickness (ALT) can be estimated from the magnitude of surface subsidence during the thaw season using simplified physical models (Liu et al., 2012, 2014, 2015; Schaefer et al., 2015; Hu et al., 2018). ALT is expected to increase as Arctic temperatures rise and permafrost undergoes long-term thaw, releasing carbon dioxide and methane, both powerful greenhouse gases. The process thus represents a potentially powerful positive feedback in the global climate system (e.g., Schaefer et al., 2009; Turetsky et al., 2020). On the other hand the active layer can also moderate the impact of surface temperature changes on deeper permafrost (Dobinski, 2011), perhaps limiting rapid increases in ALT. Frequent monitoring of ALT across the Arctic landscape is clearly important, implying the need for remote sensing approaches.

In the last three decades satellite-based Interferometric Synthetic Aperture Radar (InSAR) has been used to monitor a variety of Earth processes that generate subtle surface displacements, including earthquake and volcano deformation, and reservoir compaction from fluid withdrawal (e.g., Bürgmann et al., 2000). Recent examples include earthquake after-slip (e.g.,

Sadeghi Chorsi et al., 2022b, a), volcano deformation (e.g., Poland and Zebker, 2022; Grapenthin et al., 2022), groundwater extraction (e.g., Castellazzi et al., 2016), carbon sequestration (e.g., Yang et al., 2015; Vasco et al., 2020), seismicity induced by fluid injection (e.g., Deng et al., 2020), coastal sea ice dynamics (e.g., Dammann et al., 2019), glacier velocity estimation (e.g., Strozzi et al., 2020), and coastal flood hazard (e.g., Bekaert et al., 2017; Zhang et al., 2022). Pioneering work by L. Liu (Liu et al., 2010, 2012) demonstrated the utility of InSAR to monitor long-term permafrost thaw and changes in ALT.

Here, we use InSAR from the Sentinel-1 satellite constellation to investigate permafrost thaw on part of the North Slope of Alaska for the period 2017 to 2022, focusing on multi-year changes in ALT, and using available ICESat-2 LiDAR data to validate the InSAR result. Our study has the following objectives: (1) examine the spatial distribution of seasonal thaw subsidence amplitude using SAR interferometry from 2017 to 2022; (2) use the annual variation of InSAR-measured displacements to estimate ALT and compare to long-term in-situ ALT observations; (3) assess the ability of the ICESat-2 ATL08 product to

complement InSAR data in permafrost regions; and (4) test the influence of some environmental factors on the yearly variation of ALT.

## 2   Previous Work

Satellite remote sensing of permafrost has been ongoing for at least three decades (e.g., Peddle and Franklin, 1993), focusing on landslides and wildfires, thermokarst processes, and soil moisture dynamics. The synthetic aperture radar (SAR) data used

in many of these studies comes satellites that include Radarsat-1, Envisat, JERS-1, ERS-1 and -2, ALOS PALSAR, TanDEM-X, COSMO-Skymed, TerraSAR-X, Envisat, and Sentinel-1, and airborne sensors including UAVSAR, and AirMOSS. GNSS has also been exploited using the GPS-IR (GPS Interferometric Reflectometry) technique. These studies have been conducted the Northern and Central Alaska, Northern and Western Canada, Greenland, Antarctica, Russia, and Tibet, contributed to our understanding of permafrost dynamics and providing insights into seasonal and long-term changes in permafrost regions. Table-

1 summarizes these microwave-based studies, categorizing them into three main technical applications: 1) seasonal thaw, 2) seasonal and long-term subsidence, and 3) ALT estimation and other scientific applications. Papers most relevant to this study include Liu et al. (2010, 2012, 2015), and Schaefer et al. (2015).

## 3   Study Area

The Alaskan North Slope is bounded by the Brooks Range to the south and southeast and the Arctic Ocean to the north. Our

main study site on the North Slope is a 15 km by 30 km area in the vicinity of the Sag river and Dalton highway (69.68 N, 148.7 W; Figure 1). It is ∼50 km south of Prudhoe Bay and ∼130 km north of the Brooks Range, in the continuous permafrost region of Alaska with more than 90% permafrost coverage (Jorgenson et al., 2008).

Our study site includes Circumpolar Active Layer Monitoring (CALM) site U8. The CALM program is designed to monitor the active layer and permafrost sensitivity to climate change over extended periods, typically spanning multiple decades (Brown

et al., 2000). CALM site U8 has recorded ALT since 1996. The site encompasses a one-hectare area containing 121 sample

**Table 1.** Microwave-based studies on permafrost monitoring.

| Technical Application | Studies | Scientific Focus | Data | Study Area |
|---|---|---|---|---|
| Seasonal Thaw | Singhroy et al. (2007) | Landslide and wildfire | Radarsat-1 | Mackenzie Valley, Canada |
| | Rykhus and Lu (2008) | - | JERS-1 | Alaskan Arctic coastal plain |
| | Liu et al. (2010) | - | ERS-1 and ERS-2 | North slope of Alaska |
| | Iwahana et al. (2016) | Thermokarst, wildfire | ALOS PALSAR, GPS | North slope of Alaska |
| | Strozzi et al. (2018) | - | Sentinel-1 | Multiple sites in Alaska, Greenland, Russia and Antarctica |
| | Zwieback et al. (2018) | Thermokarst | TanDEM-X | Tuktoyaktuk coastlands, Canada and Lena River delta, Russia |
| | Bartsch et al. (2019) | - | Sentinel-1 and COSMO-Skymed | Yamal, Russia |
| | Wang et al. (2020) | - | Sentinel-1, TerraSAR-X, ALOS PALSAR | Northern Canada |
| Seasonal and Long-term subsidence | Liu et al. (2012) | - | ERS-1 and ERS-2 | North slope of Alaska |
| | Daout et al. (2017) | - | Envisat | Northwestern Tibet |
| | Chen et al. (2018) | - | Sentinel-1 | Yedoma, Russia |
| | Liu and Larson (2018) | - | GPS-IR | Barrow, Alaska |
| | Hu et al. (2018) | - | GPS-IR | Barrow, Alaska |
| | Michaelides et al. (2019) | Wildfire | ALOS PALSAR | Yukon–Kuskokwim Delta, Alaska |
| | Chen et al. (2020) | Soil Moisture | ALOS PALSAR | Toolik, Alaska |
| | Bernhard et al. (2020) | Thermokarst | TanDEM-X | Northern Canada |
| | Honglei et al. (2021) | - | ALOS PALSAR | Qinghai-Tibet Plateau |
| ALT estimation; other | Liu et al. (2012) | - | ERS-1 and ERS-2 | North slope of Alaska |
| | Schaefer et al. (2015) | Thermokarst | ALOS PALSAR | Barrow, Alaska |
| | Michaelides et al. (2019) | Wildfire | ALOS PALSAR | Yukon–Kuskokwim Delta, Alaska |
| | Chen et al. (2020) | Soil Moisture | ALOS PALSAR | Toolik, Alaska |
| | Michaelides et al. (2021b) | Soil Moisture | L-band UAVSAR and AirMOSS P-band | Alaska and western Canada |
| | Chen et al. (2023) | Soil Moisture | L-band UAVSAR and AirMOSS P-band | Alaska and western Canada |

square arrays, each measuring approximately 10 m horizontally. It is located 88 m above sea level, is relatively flat, and lies within an inner coastal plain with river terraces. The site has an organic layer ∼23 cm thick that is usually water-saturated during thaw season. U8's vegetation coverage is classified as graminoid-moss tundra, graminoid prostrate-dwarf-shrub, and moss tundra. Its soil texture is classified as predominantly sand, gravel and peat. Soil taxonomy is Ruptic-Histic-Aquorthel
(Ping et al., 2015; Staff, 1999), i.e., a poorly drained, occasionally to frequently water-saturated soil with a significant amount of organic matter (https://www2.gwu.edu/~calm/data/webforms/u8_f.htm). A 12 km by 12 km test site around U8 (red box in Figure 1) is used for focused studies of ALT estimation, based on our InSAR-derived displacement estimates during thaw season.

The Circumpolar Arctic Vegetation Map (CAVM) at this location describes graminoid and prostrate-dwarf-shrub vegetation
5-10 cm in height. This vegetation structure is favorable for shorter radar wavelength radars such as Sentinel-1's C-band (∼5.5 cm wavelength) to retain phase coherence, but also suggests that accounting for vegetation height will be important to assess seasonal and longer term elevation changes in this area.

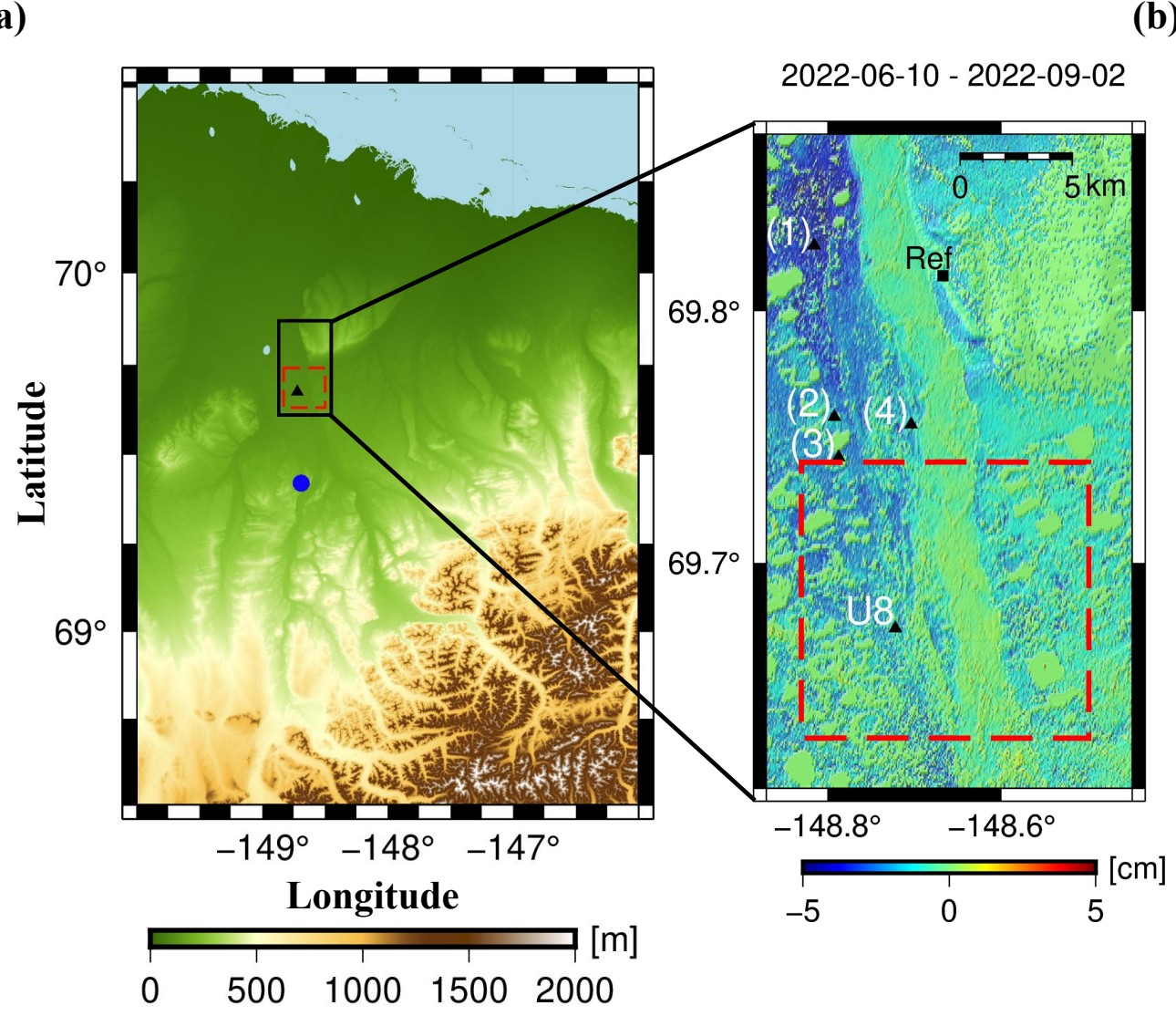

**Figure 1.** (a): DEM including study area (black box) in Northern Alaska. Black box is expanded in Figure 1b, red boxes outline focused test area shown in Figure 5. Black triangle shows CALM site (U8) where ground-based measurements of ALT are available. Blue circle represents location of closest meteorological station (Sagwon). (b): Line of sight (LOS) displacement of the study area from 2022-06-10 to 2022-09-02 as measured by InSAR. Negative values mean displacement away from satellite, positive values mean displacement towards satellite. DEM relief map is shown in background. Triangles show location of CALM site and displacement time-series shown in Figure 3. Black square represents reference point used for InSAR analysis.

## 4  Methods

### 4.1  InSAR Data Processing

#### 4.1.1  Data and Material

We used the Alaska Satellite Facility's Hybrid Pluggable Processing Pipeline (HyP3) software to form interferograms from Sentinel-1 SAR data (Hogenson et al., 2020). HyP3 uses the Copernicus GLO-30 Digital Elevation Model (DEM) for scene co-registration and topographic phase corrections (ESA, 2021). Interferograms were filtered using the adaptive phase filter in Goldstein and Werner (1998). Individual interferograms were unwrapped using a minimum-cost-flow algorithm (Chen and Zebker, 2002) and geocoded to a 30 m grid spacing. We used the open-source Miami InSAR time-series software in python (MintPy) to generate LOS displacement time-series from the unwrapped and geocoded interferograms (Fattahi et al., 2016; Yunjun et al., 2019). Interferograms with high spatial coherence and short time-interval between scenes were chosen to avoid decorrelation and phase unwrapping errors. Phase unwrapping artifacts occur in permafrost regions when disconnected wetlands and large seasonal deformation preclude smooth unwrapping of the phase (e.g., Strozzi et al., 2018). Noisy interferograms were removed from the time-series and seasonal amplitude inversion processes (Section 4.3.2). Geocoded LOS displacement data for active layer thickness estimation was then extracted for the study area.

Significant changes in scattering characteristics are expected during the freeze season when the surface is covered with snow and ice, hence we focus on the summer thaw season. Variations in soil moisture can also significantly affect coherence. To mitigate this possibility, we looked for noisy interferograms, which could be partly due to such moisture changes, and excluded these from our analysis. We employed two criteria to assess noise. First, we manually reviewed the interferograms and eliminated noisy ones. Second, we assessed the spatial coherence of the chosen interferograms to ensure they all had high coherence. Note that data were collected at a time likely to retain soil moisture, the period that begins immediately after snowmelt, when the soil column is saturated, and end near the end of thaw season, when the soil column may still be wet due to the complete thawing of any residual ice in the active layer. We focused our study on the June to September time frame, using Sentinel-1 SAR images with a 12 day revisit interval and descending geometry for the years 2017 to 2022 (Table S1). Available meteorological data suggest no anomalous drought periods during these years.

#### 4.1.2  Reference Point Selection

InSAR measures phase differences between SAR observations in space and time. To relate these phase difference measurements to surface displacement, a reference location with assumed or known displacement is required, with high temporal coherence ($> 0.8$) to avoid introducing noise into the time series. In most permafrost regions, rock outcrops are a good reference as they can be assumed to show only minimal displacement. However, they may not be available for all regions. Liu et al. (2010) point out that river floodplains usually have well-drained sandy soils and hence tend not to experience significant frost heave. They may be used as reference points if they are not within a river channel, which can undergo large elevation changes

from erosion/deposition events (see Figure S3). Figure 1b shows our reference point, a rock outcrop which remains coherent (temporal coherence ~0.95) during the 2017 to 2022 thaw seasons.

### 4.1.3 Atmospheric Delay Correction

Atmospheric effects are one of the main error sources in the InSAR process (Meyer et al., 2006). While InSAR data can be affected by both the ionosphere and the troposphere, here we focus on tropospheric effects as ionospheric impacts are less pronounced in C-band data (Meyer, 2011). Tropospheric phase impacts can be modeled as (Ding et al., 2008):

$$\Delta\phi = \phi_2 - \phi_1 = \frac{4\pi}{\lambda}[d_2 - d_1] + \frac{4\pi}{\lambda}[\delta d_2 - \delta d_1] \tag{1}$$

where $\phi$ is the phase of a SAR image, $d$ is the range from satellite to surface, $\delta d$ is the tropospheric propagation delay, and $\lambda$ is the radar wavelength. Tropospheric phase signals in InSAR data can be caused by two processes: changes in the atmospheric stratification and turbulent mixing. The stratified component typically correlates with topography (Hanssen, 2001) and may be estimated and then removed based on delay-elevation correlations (Doin et al., 2009). The turbulent component is usually much less than the stratified component, but is uncorrelated in time and space and hence harder to predict or measure. According to (1) if the atmospheric propagation conditions at the time of SAR acquisitions are not the same ($\delta d_2$- $\delta d_1 \neq 0$), then tropospheric phase components will be introduced, contaminating the true displacement signal. Applying atmospheric corrections to C-band radar images can improve signal to noise ratio, especially when there is a considerable height difference between the study area and reference point. We applied the atmospheric correction model described in Jolivet et al. (2011, 2014) using ECMWF reanalysis (ERA-5) datasets (Hersbach et al., 2020). This approach mainly reduces the stratification component of the tropospheric delay.

### 4.2 ICESat-2 Data Processing

To validate our InSAR measurements of thaw season subsidence, we used independent LiDAR elevation data from the ICESat-2 satellite (Martino et al., 2019). The ATLAS (Advanced Topographic Laser Altimeter System) LiDAR on ICESat-2 uses a multi-beam photon-counting laser operating at 532 nm, i.e., the green portion of the electromagnetic spectrum. Surface range is determined by the travel time of each detected photon. When coupled with the satellite's position, the range data provides accurate geolocation of the surface, in this case referenced to the WGS-84 ellipsoid. With a laser repetition rate of 10 kHz, pulses occur approximately every 70 cm on Earth's surface. Each footprint is about 13 m in diameter. Beam pairs, with different energies to adjust for surface reflectance, are spaced about 3.3 km apart across tracks, forming six tracks with beams in each pair separated by 90 m. Ranging precision for flat surfaces is expected to have a standard deviation of around 25 cm, primarily influenced by ATLAS timing uncertainty (Neuenschwander et al., 2019).

The ATL08 product algorithm is designed to extract terrain and canopy heights from vegetated surfaces using the geolocated photons (Neuenschwander et al., 2019). We used the "h_te_best_fit" parameter, which estimates terrain height by fitting a plane to along-track points in each 100 m segment and reports the height of the middle of the fitted plane (Neuenschwander

and Magruder, 2019; Neuenschwander et al., 2021), reducing the impact of random errors. The height of the terrain midpoint is calculated by choosing the best fit among three models: linear, third-order, and fourth-order polynomials applied to the terrain photons. This allows for interpolation of the elevation at the midpoint of the 100 m segment (Neuenschwander and Magruder, 2019; Neuenschwander et al., 2021; Neuenschwander and Pitts, 2019). The standard deviation of terrain points around the interpolated ground surface within the segment is one measure of surface roughness. Neuenschwander and Pitts (2019) provide additional details describing the ATL08 algorithms. ATL06 is an alternate product algorithm, optimized for ice surfaces, and has been used in some permafrost studies (e.g., Michaelides et al., 2021a) (See Supplement).

While the nominal temporal resolution of ICESat-2 data is 91 days, cloud cover often limits the amount of usable data in Alaska (e.g., Neuenschwander and Pitts, 2019). Two repeat track observations were available in our study area, acquired on 2021-06-08 and 2021-09-06. Due to pointing-related uncertainty, observations are not always repeated in expected locations, which amplifies the height uncertainty. To address this issue, we divided the study area into 50 m grid cells, and assigned each observation in 2021-06-08 and 2021-09-06 repeat tracks to one of those grid cells. Figure S3 shows the height difference between all reported terrain observations (ATL08) in the study area with same location between 2021-06-08 and 2021-09-06. In the limited area where both InSAR and ICESat-2 data are available, we used the ICESat-2 data to compare to our InSAR results (Figure 3 and Supplement).

## 4.3 Active Layer Thickness Estimation Model

To relate the InSAR observations to ALT, we assume that the measured LOS displacements are predominantly due to vertical motion (negligible horizontal motion) and that this vertical motion is caused by thawing ground ice in the active layer. The assumption of negligible horizontal motion is justified because over the short data time interval the technique is not sensitive to long-term tectonic motion. Displacements from the M 6.4 August 2018 earthquake, ∼130 km east of study area (USGS hypocenter at 145.291 W, 69.576 N, depth 15.8 km) are negligible. Most surface motion in the thaw season therefore likely reflects thawing ground ice. We project the LOS displacements into the vertical direction using the local incidence angle ($\theta$) for each radar pixel (see Equation 3). We follow the simplified Stefan solution to estimate depth of thawing in the soil (Nelson et al., 1997) aided by field-observed air temperature data. We also assume that subsidence can be related to a simple thaw index, for example the accumulated degree days of thawing (ADDT). Our procedures are virtually identical to those described in Liu et al. (2012) and Schaefer et al. (2015), with the exception that we do not estimate multi-year subsidence and do not average ALT across multiple thaw seasons. This enables us to directly compare our space-based estimates with yearly ground truth estimates from CALM site U8 and evaluate inter-annual changes in ALT.

### 4.3.1 Accumulated Degree Days of Thawing Calculation

To calculate ADDT, we use the NOAA Climate Data Online (CDO) tool to find nearby meteorological stations. The closest station is ∼30 km south to our test area (Name: Sagwon, Figure 1a). We assume that our test area has the same temperature trend as this station for the 2017 to 2022 thaw seasons. We define the first and last day with temperature > 0 °C as the first and last day of the thaw season. ADDT is defined by the following equation (Riseborough, 2003):

$$ADDT = \int_0^{\alpha_s} (T_s - T_f)dt \approx \sum_0^{\alpha_s} \bar{T}_s \tag{2}$$

where $\alpha_s$ is the duration of thawing season, in days. $T_s$ is surface temperature (°C), $T_f$ is equal to freezing point, 0 °C, and $\bar{T}_s$ is daily mean surface temperature. Due to lack of in-situ surface temperature data, we set $\bar{T}_s$ using air temperature observations.

### 4.3.2 Seasonal Amplitude Inversion

The relationship between the seasonal vertical surface displacement magnitude and ADDT can be written as (Liu et al., 2012; Schaefer et al., 2015):

$$D_i = \frac{LOS}{\cos(\theta)} = E(\sqrt{A_{2,i}} - \sqrt{A_{1,i}}) + \varepsilon \tag{3}$$

where $D_i$ is the vertical displacement estimate for a given pixel in the $i^{th}$ interferogram, $\theta$ is the local incidence angle at that pixel calculated from nadir, $E$ is the amplitude of the seasonal vertical displacement estimate which reflects physical parameters such as soil thermal conductivity, latent heat of fusion, soil density and relative water content (Nelson et al., 1997). $A_{1,i}$ and $A_{2,i}$ are normalized accumulated degree days of thawing at the first and second acquisition date. $\varepsilon$ is an error term that captures model deficiencies, noise, and other unknown error sources. We do not consider secular (long-term) displacement signals in (3) because we analyze the thaw seasons of 2017 to 2022 separately. This is the major difference between our approach and those described in Liu et al. (2012), Schaefer et al. (2015) and Michaelides et al. (2019), where seasonal and interannual trends were estimated simultaneously.

We can rewrite (3) in matrix form, considering the interferograms listed in table S1, to estimate $E$ using least squares for separate thaw seasons:

$$\begin{bmatrix} D_1 \\ D_2 \\ \vdots \\ D_N \end{bmatrix} = \begin{bmatrix} \sqrt{A_{2,1}} - \sqrt{A_{1,1}} \\ \sqrt{A_{2,2}} - \sqrt{A_{1,2}} \\ \vdots \\ \sqrt{A_{2,N}} - \sqrt{A_{1,N}} \end{bmatrix} \begin{bmatrix} E \end{bmatrix} \tag{4}$$

### 4.3.3 Active Layer Thickness Inversion

If we assume that the seasonal vertical surface displacement amplitude $E$ is caused only by thawing ice and corresponding volume reduction, we can write $E$ as a function of physical properties such as soil porosity, soil moisture fraction, and density of ice and water through a vertical profile from surface to depth of the active layer (Liu et al., 2012; Schaefer et al., 2015):

$$E = \frac{\rho_w - \rho_i}{\rho_i} \int_0^{ALT} P(z)S(z)dz \tag{5}$$

The variables $\rho_w$ and $\rho_i$ in (5) are the density of water and ice [$\mathrm{kg\,m^{-3}}$], respectively. $P(z)$ is the soil porosity which is a function of depth and depends on soil content, and $S(z)$ is the soil moisture fraction of saturation. Following Schaefer et al. (2015), we assume $S(z) = 1$, which means that the active layer is fully saturated and saturation does not change with depth.

### 4.3.4 Porosity Model

Following earlier authors, we assume soil in the active layer consists of organic matter and mineral soil, with porosity decreasing exponentially with depth due to decreasing organic matter. There is one active CALM site in our test area: U8 (Figure 1b). This site is described as having 23 cm organic layer thickness, consisting mainly of peat plus sand and gravel (Section 3). We applied the formulation introduced by Liu et al. (2012) and assume that is the weighted average of organic and mineral matter:

$$P = f_{org} P_{org} + (1 - f_{org}) P_{min} \tag{6}$$

where $P$ is the porosity, and $f_{org}$ is defined as the organic soil fraction by Schaefer et al. (2009) as:

$$f_{org} = \frac{M_{org}}{M_{org\_max}} = \frac{\rho_{org}}{\rho_{org\_max}} \tag{7}$$

In (7), $M_{org}$ and $\rho_{org}$ are the simulated mass of organic matter and organic soil density in a given layer of soil, respectively. $M_{org\_max}$ and $\rho_{org\_max}$ are bulk organic matter mass and bulk density for pure organic soil, respectively. We set $P_{org} = 0.95$

based on the model from Bakian-Dogaheh et al. (2022). The porosity of mineral soil then depends on the sand fraction of soil. To estimate $P_{min}$ we used the porosity-sand fraction relation provided in Liu et al. (2012):

$$P_{min} = 0.489 - 0.00126 \, fr_{sand} \tag{8}$$

We used Global Land Data Assimilation System (GLDAS) soil parameters with $0.25°$ spatial resolution to extract the soil sand fraction (Rodell et al., 2004). We set $P_{min} = 0.488$, and $\rho_{org\_max} = 130$ [$\mathrm{kg\,m^{-3}}$] for bulk density of peat (Grigal et al.,

1989; Hossain et al., 2015). As mentioned earlier, to formalize with depth, we assume the organic matter amount decreases exponentially with depth:

$$\rho_{org} = B \exp(-kz) \tag{9}$$

where $k$ is an empirical constant [$\mathrm{m^{-1}}$], set to 5.5 (Liu et al., 2012; Jackson et al., 2003). To retrieve $B$, we use simulated mass of organic matter ($M_{org}$: total soil carbon content) from Johnson et al. (2011) and Mishra and Riley (2012) and ensure

that total carbon mass is conserved:

$$\int_0^{root} B \exp(-kz) \, dz = M_{org} \tag{10}$$

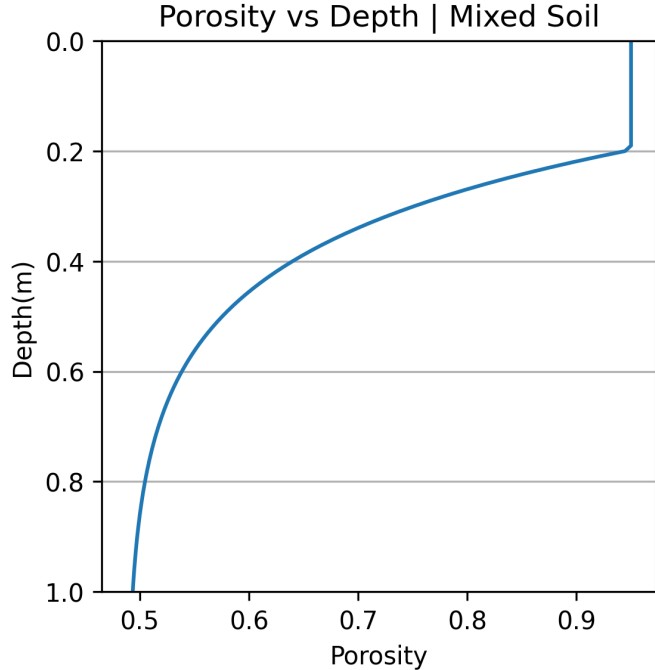

**Figure 2.** Depth-porosity model used in this study assuming a mixture of organic and mineral matter.

We set $M_{org}$ = 70 [$\mathrm{kg\,m^{-2}}$] (Johnson et al., 2011; Mishra and Riley, 2012). The spatial divergence of total soil carbon content for the 0-100 cm depth range is large in Arctic tundra regions considering vegetation type. Mishra and Riley (2012) and Johnson et al. (2011) estimate total soil carbon around our study area at [60- 80] and [50-70] [$\mathrm{kg\,m^{-2}}$], respectively. Root depth is the maximum observed ALT at a given site since roots cannot penetrate solid ice. Here, we set maximum root depth at 1.1 m because maximum observed ALT at site U8 is reported as $\sim$1.1 m for 2022. Then we solve (10) for $B$ and replace it in (9). Figure 2 shows the relation between porosity and depth in a mixed soil. We set $P = 0.95$ for the first $\sim$23 cm depth reflecting organic matter thickness. After 23 cm depth, the porosity decreases exponentially, reaching its minimum near the top of the frost table. Finally, we put all equations into (5) and use a numerical bisection algorithm to solve for the upper integral limit, ALT. We set the accuracy of bisection to be at the mm level.

## 5  Results and Discussion

### 5.1  Estimating Seasonal Vertical Displacement

Figure 3 shows displacement time-series for the four test locations shown in Figure 1b. All four locations show subsidence during thaw season. The maximum amplitude of subsidence ranges from 20 mm (Location 4) to 60 mm (Location 1). In 2021, the subsidence amplitude was small and similar among the four locations ($\sim$20 mm). The subsidence rate is approximately

constant during the thaw season. The root mean square error (RMSE) of the linear fit to the displacement data is less than 7 mm at all four locations over all six years. The maximum subsidence rate observed during the short ($\sim$3-4 month) thaw season is $\sim$18 mm/month.

Note that we do not attempt to connect our displacement time series across adjacent years. The freezing process, consequent frost heave, and deep snow at these sites during winter makes phase connection difficult due to loss of coherence (e.g., Strozzi et al., 2018). Nevertheless, our approach can still be used to assess long term (multi-year) changes in permafrost, as shown in Section 5.3.

Figure 4 shows the seasonal subsidence rates at the four locations over the six-year test period. No clear long-term trend is observed. Location (1) has the largest rate variation, from 4 mm/month in 2018 to 18 mm/month in 2020. Location (3) has the minimum rate variation, 5 mm/month in 2017 to $\sim$10 mm/month in 2021. We do not observe spatial correlation between subsidence rates at the various locations. For example, location (1) shows the fastest subsidence, with high rates in 2017, 2020 and 2022 but much smaller rates in 2018, 2019 and 2021. Location (2)'s fastest subsidence occurs in 2019 while the fastest rates for location (3) and location (4) occurred in 2019 and 2021.

### 5.2 Validation of InSAR Surface Displacement Estimates with ICESat-2 Data

We used the ICESat-2 ATL08 data product to compare with our InSAR time-series of relative height change for the 2021 thaw season (Figure 3). Comparisons using the optical LiDAR data are primarily limited by cloud cover, however all four of our test locations had suitable ICESat-2 data at the beginning and end of the 2021 thaw season. To minimize the effect of systematic errors, we used repeat data from the same reference ground track (RGT=1150), and considered only elevation change during thaw season, referencing the height of the second acquisition (end of thaw season; 2021/09/06) to the first (beginning of thaw season; 2021/06/08). The height of the first date's LiDAR data is assigned 'zero elevation', to agree with the InSAR estimate. This is a reasonable assumption because the two datasets have similar start dates in 2021, June 8 for ICESat-2, and June 3 for SAR.

Figure 3 shows that with these assumptions, the LiDAR and radar approaches agree well. Since the two approaches to elevation change estimation are independent, their agreement is a strong validation of the InSAR approach, albeit limited to a small number of test cases. The agreement between the two approaches also suggests that our reference point for the InSAR data experiences negligible change during the study period. Reference point selection for InSAR is difficult in remote permafrost regions as most areas undergo subsidence during the thaw season. ICESat-2 data when available could help in this regard, but will be limited by pointing errors, cloud cover, and density of the vegetation canopy.

We also tested the utility of ICESat-2's ATL06 data product ("h_li") (Smith et al., 2019), described in the Supplement. For this test, we expanded the comparison to a larger number of sites and dates, but otherwise used the same general procedures as for the ATL08 product. For our original four test sites, the ATL08 product shows better agreement with the InSAR data (Figure S4-S5). In our larger study area, 78 cells reported both ATL08 and ATL06 data. For these cases, the two products are equivalent (within 1 cm height difference) in 9% of cases, and agree within 10 cm in 61% of cases (Figure S4). Including the four original test sites described above, for the 15 cases where InSAR and ICESat-2 products were available, ATL08 shows

better agreement with InSAR in 8 cases, while ATL06 shows better agreement in 7 cases. In relatively flat areas, both data products show similar performance. The larger footprint of the averaged ATL08 data product (100 m) compared to ATL06 (40 m) may also be advantageous given the high spatial variability of ALT (see section 5.3). The Supplement also includes a description of the formal uncertainties associated with both the ATL06 and ATL08 data products.

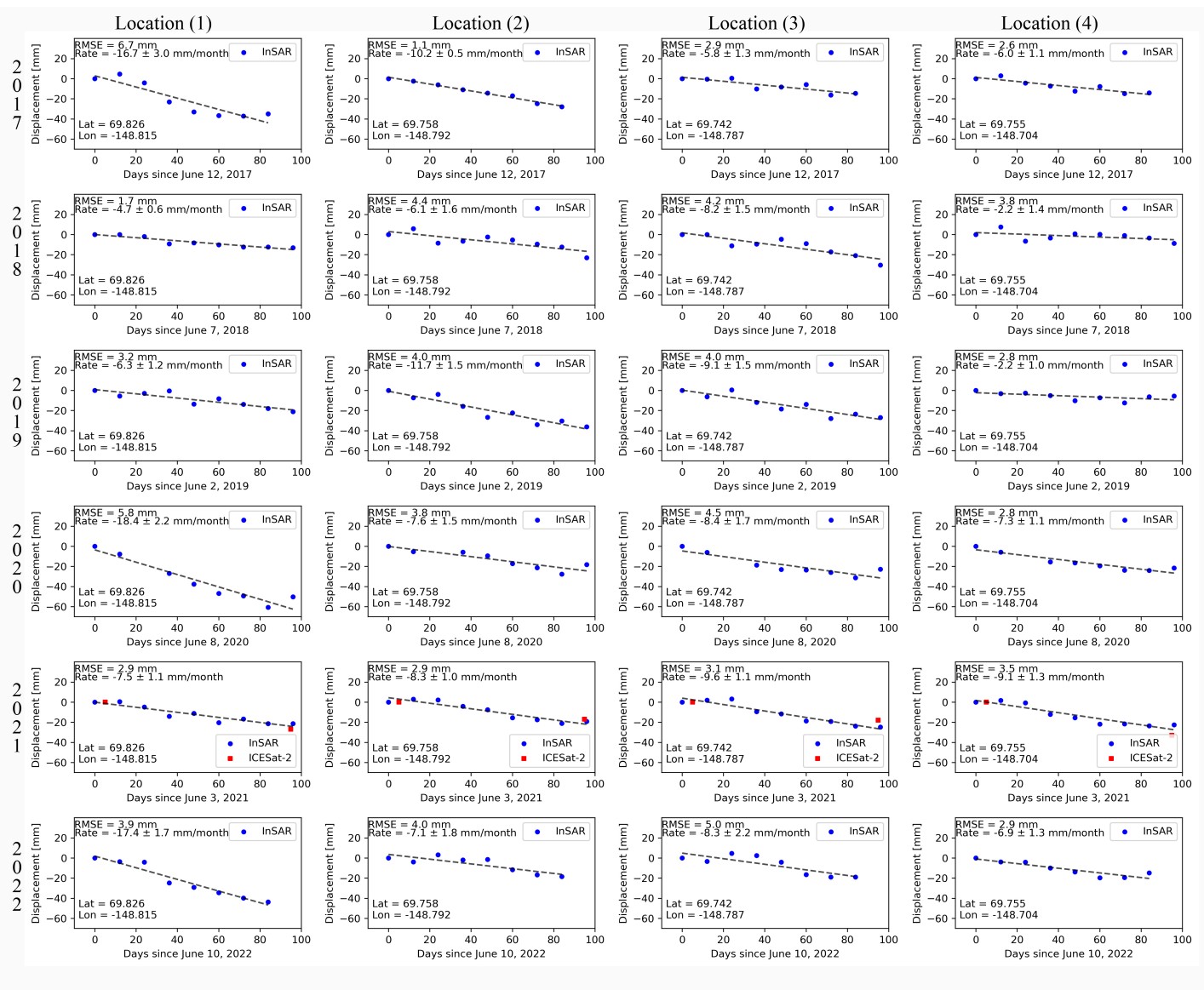

**Figure 3.** LOS displacement time-series for four test locations (black triangles in Figure 1b) with respect to the first SAR acquisition in the thaw season. Black dashed lines are best-fitting regression lines for InSAR LOS displacement only. Rate and RMSE of fitted lines are shown in the top-left of each sub-figure. Red squares in 2021 show ICESat-2 ATL08 terrain height product. Latitude and longitude of each analyzed location are shown in the bottom-left of each panel. Note that subsidence only occurs during the thaw season and not the entire year.

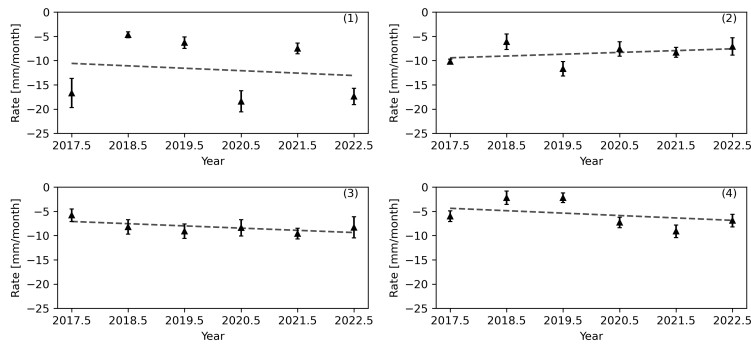

**Figure 4.** Rate comparison of LOS displacement during the summer thaw season for the locations shown in Figure 1b. Rate and an error bar are from fitted linear line (See Figure 3).

**Table 2.** Thaw onset, end and ADDT for 2017 to 2022 based on Sagwon station (Figure 1).

| Year | Thaw onset [m / d] | Thaw end [m / d] | ADDT [$^\circ$C day] |
|------|--------------------|------------------|----------------------|
| 2017 | 5/24 | 9/19 | 980.8 |
| 2018 | 6/12 | 9.21 | 713.5 |
| 2019 | 5/20 | 9/14 | 1040.2 |
| 2020 | 5/21 | 9/19 | 875.7 |
| 2021 | 5/21 | 9/17 | 943.5 |
| 2022 | 5/19 | 9/21 | 970.8 |

### 5.3   Active Layer Thickness Estimation and Validation

Figure 5 shows the seasonal vertical displacement amplitude and its RMSE calculated from equation (4) and estimated ALT from 2017 to 2022 in our test area (red box in Figure 1) using procedures described in Section 4.3. Minimum ALT occurred in 2018 and 2021. Maximum ALT occurred in 2019 and 2020. The variation in these estimates may in part reflect uncertainty in thaw season length. Thaw season usually starts around May 20 and ends around September 20, but the accumulated degree days of thawing differ each year. Sagwon station data for this time period shows that maximum and minimum ADDT occurred

in 2019 and 2018, respectively (Table 2). The overall pattern of ALT remained the same in 2017, 2018, 2021 and 2022, but differed in 2019 and 2020. Maximum ALT occurred west and west-south of site U8. This was also true in 2019 and 2020 but spatial variation was higher than other years.

ALT in some areas showed a deeper-than-usual pattern in 2019 and 2020, but recovered in 2021 and 2022. For example, an area a few kilometers south-east of U8 showed high variability in 2019 and 2020, but shallow ALT before (2017 and 2018)

and after (2021 and 2022). Deeper ALT in 2019 correlates with ADDT. However, deeper ALT in 2020 and shallower ALT in other years does not clearly correlate with ADDT — 2020 was the second coolest thaw season in our study period with ADDT = $\sim$876 $^\circ$C day (Table 2).

We can compare our results with in-situ data. CALM site U8 is a one hectare area with 121 samples in a square array. Each sample area is 10x10 m. Its ALT has been observed at the end of the thaw season since 1996. Thaw depth is measured by pushing a metal rod into the soil to refusal, assumed to represent the top of the permanently frozen layer. ALT is not reported if the probe intersects ponded water or rocks. The mean of all 121 ALT measurements and the corresponding RMSE are reported. Our approach for reporting InSAR-derived ALT is similar. The pixel to closest to U8 and adjacent pixels within 50 m in the east-west and north-south directions are defined. We report the mean of these pixels and their RMSE for comparison with in-situ data.

Figure 6 shows ALT data around the U8 CALM site for different years. Our estimated ALT agrees within uncertainty with the in-situ data for all five of the years when data is available. In-situ ALT is not reported for 2020. This agreement suggest that our assumptions about model parameters, based on available in-situ data and published literature, are reasonable.

Chen et al. (2020) estimated ALT and volumetric water content for large areas in Alaska including the U8 CALM site using L-band UAVSAR and AirMOSS P-band polarimetric SAR, respectively. Their result is in good agreement with the in-situ data and this study considering joint uncertainties. Data processing details are provided in Michaelides et al. (2021b) and Chen et al. (2023).

To assess the agreement between in-situ data and estimated ALT, we follow Liu et al. (2012) and use (11) to evaluate whether a given year's InSAR-based estimate of ALT is consistent with the in-situ observation given its data uncertainty:

$$r^2 = \left( \frac{ALT_{Mod} - ALT_{Obs}}{\sigma_{Obs}} \right)^2 \tag{11}$$

where the numerator is the residual between in-situ and InSAR-based ALT, and the denominator is the reported in-situ data uncertainty. $r^2$ values lower than 1 indicate good agreement. Except for the 2017 estimate, with $r^2 = 2$, all other years have $r^2$ less than 1. Estimated ALT in 2022 show the best agreement with $r^2 = 0.3$. Figure S2 gives more details.

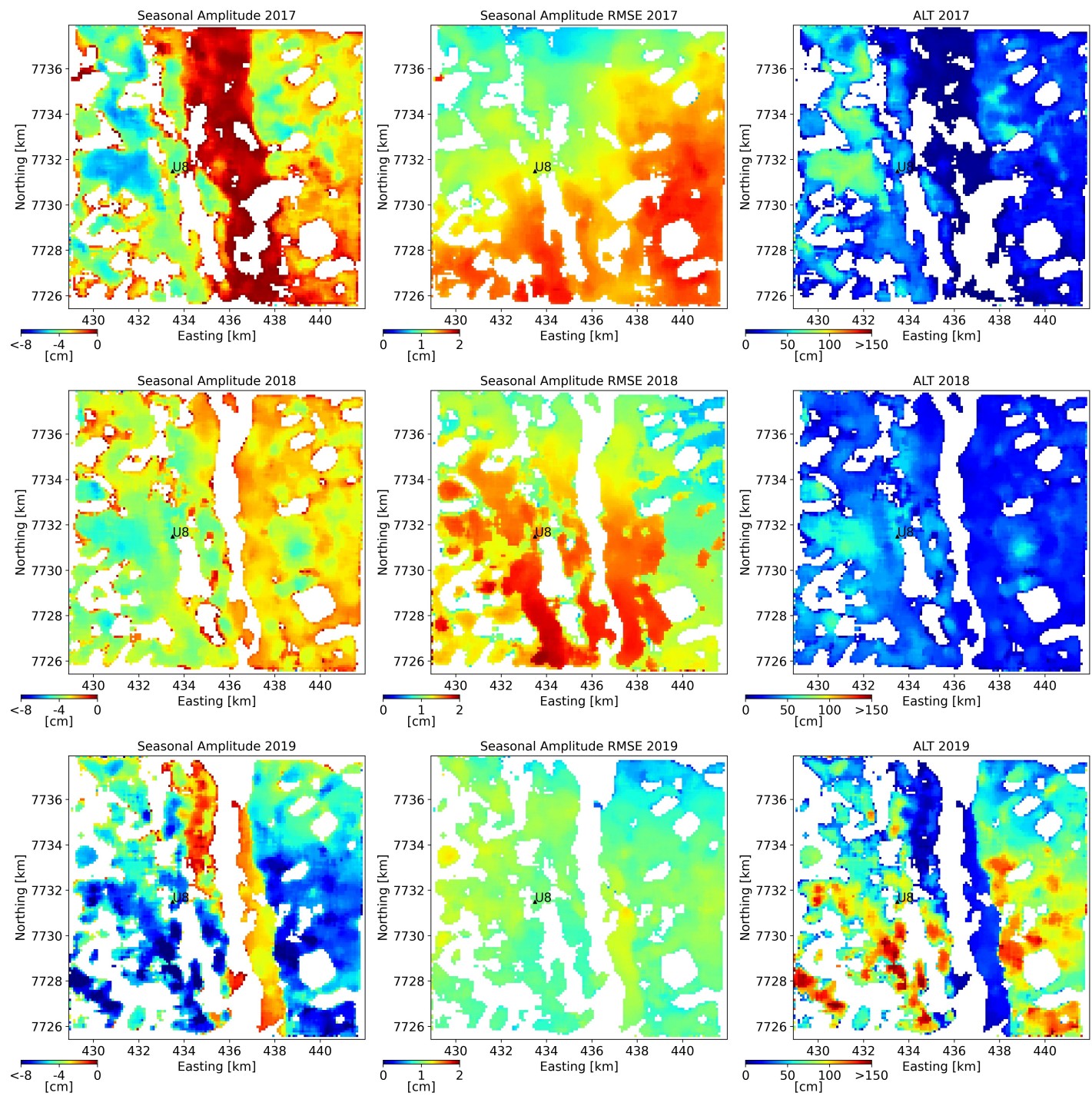

**Figure 5.** Estimated seasonal amplitude, its RMSE and ALT for study area (red box Figure 1a,b) from 2017 to 2022. Black triangle shows location of CALM site U8. White areas represent low coherence which are masked out in the model calculations. The Sag river runs south to north in the center of each panel (See Figure S3).

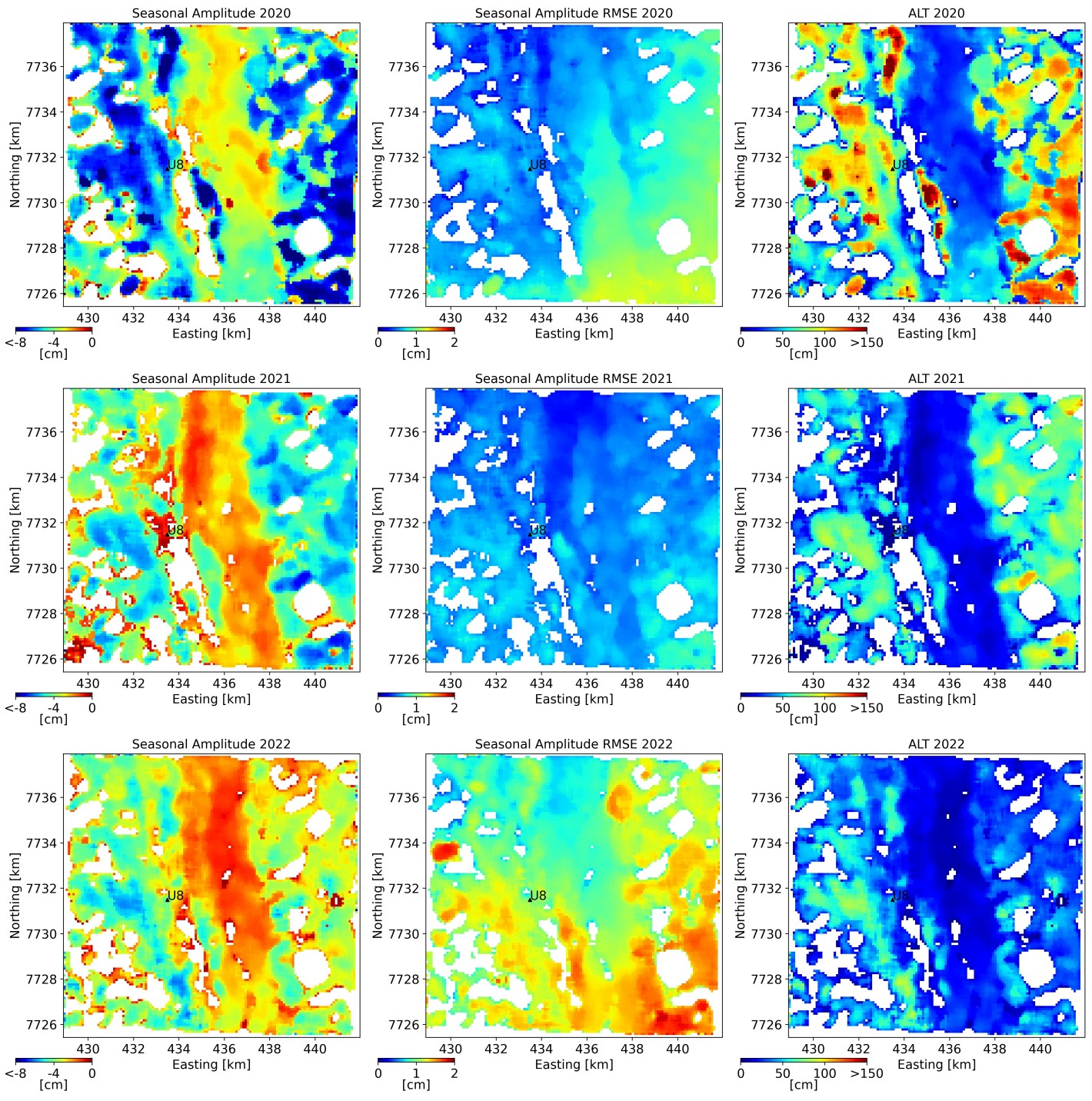

**Figure 5.** Continued

Our results and the in-situ data suggest that ALT exhibits high spatial variability. It is generally assumed that ALT depends on parameters such as ADDT, precipitation, and local topography, the latter reflecting its influence on soil moisture and aspect.

Our results show a moderate correlation with ADDT but no correlation with precipitation, although the latter could reflect limited spatial resolution of the available data.

The influence of local topography on ALT can be tested by examining available high resolution in-situ data. Data from the U8 CALM site provides an excellent opportunity to investigate both spatial and temporal variability of ALT. Over this small area we expect that local topography will show minimal year to year variation. Figure 7 shows ALT variation over an 11x11

square array of sample points, with each point sampling an area of 10x10 m. Data are available for the period between 1996 and 2022 with a gap in 2020. Location is described in local coordinates. We also show the RMSE of each grid point from its average over this time period, and a time series of ALT for three representative points in the array.

Even over this small area we see no significant spatial or temporal pattern in ALT over the quarter-century period of available data. At least for this example, the influence of local topography appears to be minimal, although we cannot preclude

microtopographic (less than 10 m) effects that vary over time.

The Mann-Kendall test was employed to evaluate this in a rigorous way. The test determines if a significant monotonic trend is present for either increasing or decreasing ALT at each grid point. Data spanning from 1996 to 2019 were analyzed due to the absence of data in 2020. To maintain consistency and account for possibly significant temporal variation in ALT, data from 2021 and 2022 were excluded. The null hypothesis was rejected for 31.4% of the cells; the remaining 68.6% of cells do not

show a statistically significant trend. In other words, only 38 out of 121 cells had a significant increasing or decreasing ALT trend. Among these 38 cells, 35 cells showed an increase in active layer thickness over the sample time period. The maximum RMSE of the cells is ∼20 cm. Variation of the same grid cell in two consecutive years reaches as high as ∼60 cm. Since air temperature (related to ADDT) and precipitation are unlikely to vary significantly over this 100x100 m area, and since overall morphology is unlikely to vary significantly over this time period, other factors must explain the variation in ALT. Given that

our estimated ALT aligns well with in-situ ALT (Figure 6, Figure S2) and that the long-term in-situ ALT measurements (2002-2022) show no correlation with ADDT and precipitation (Figure 8), we suggest that other factors are likely influencing the results. Micro-topographic effects, temporal changes in sub-surface moisture flow, soil organic content and vegetation growth and decay are possible factors. Nelson et al. (1998), Nelson et al. (1999) and Hinkel and Nelson (2003) conclude that in-situ ALT shows Markovian behavior with high spatial and temporal variation.

## 5.4 Relation of Meteorological Parameters to Active Layer Thickness

We investigated correlations between in-situ ALT and several meteorological parameters, including ADDT and precipitation in thaw seasons from 2002 to 2022. ADDT and precipitation data are from the Sagwon meteorological station. Figure 8a shows the relation between ADDT and ALT. From Stefan's equation, we expect a positive correlation between ADDT and ALT. However, the correlation is weak (R-squared = 0.42; Figure 8b) suggesting the influence of additional factors. Precipitation may influence

ALT, e.g., by advecting heat downward to promote permafrost thaw, but there are additional factors to consider. For example, an increase in soil moisture leads to a rise in the thermal conductivity of soil, potentially increasing the depth of the active

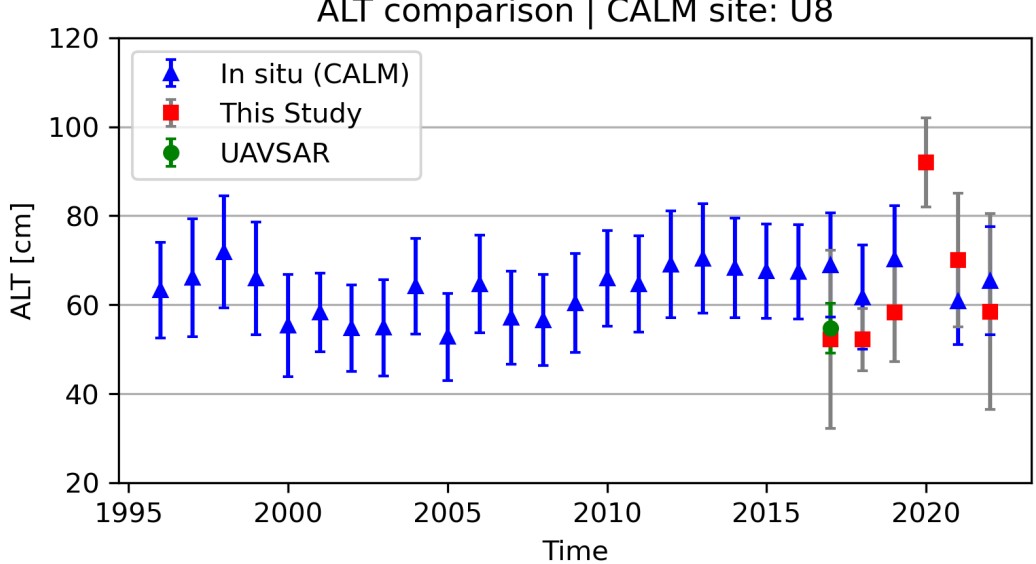

**Figure 6.** ALT comparison at CALM site U8. Blue triangles represent average in-situ ALT from manual mechanical probing across all grid cell from 1996 to 2022 (https://www2.gwu.edu/~calm/data/north.htm). Green circle is estimated ALT for the closest pixel to U8 using airborne L- and P-band SAR images (Chen et al., 2022). Red squares (this study) are average estimated ALT for pixels with 50 m of U8. In-situ ALT is not reported for 2020.

layer during thaw season. However, an increase in soil moisture also increases the total amount of heat required for thawing, promoting a shallower active layer. Clayton et al. (2021) showed that ALT has a positive correlation with volumetric water content (VWC) in the upper 12 cm of soil, a negative correlation with bulk VWC, and no statistically significant correlation with VWC in the upper 20 cm of soil. We also do not see a statistically significant correlation between ALT and precipitation, perhaps reflecting these opposing impacts (Figure 8c).

We used simple regression analyses to relate ALT to several multi-parameter factors including ADDT, precipitation and accumulated degree days of freezing (ADDF) from the previous year. However, these did not improve the correlation. Perhaps other factors such as local elevation gradients (influencing local hydrology), vegetation type, or the previous year's snowfall need to be considered. It is also possible that some of the variability in our ALT estimates reflects instead variations in total ice content (Zwieback et al., 2024).

### 5.5 Applicability to other Regions

Alaska's North Slope is an optimum region for InSAR-based approaches to permafrost monitoring because of limited tree cover. We also tested our technique in a region with more extensive tree cover, the Beta site of the APEX (Alaska Peatland EXperiment) project, located approximately 30 miles southwest of Fairbanks (64.696 N, 148.322 W). This site is located in

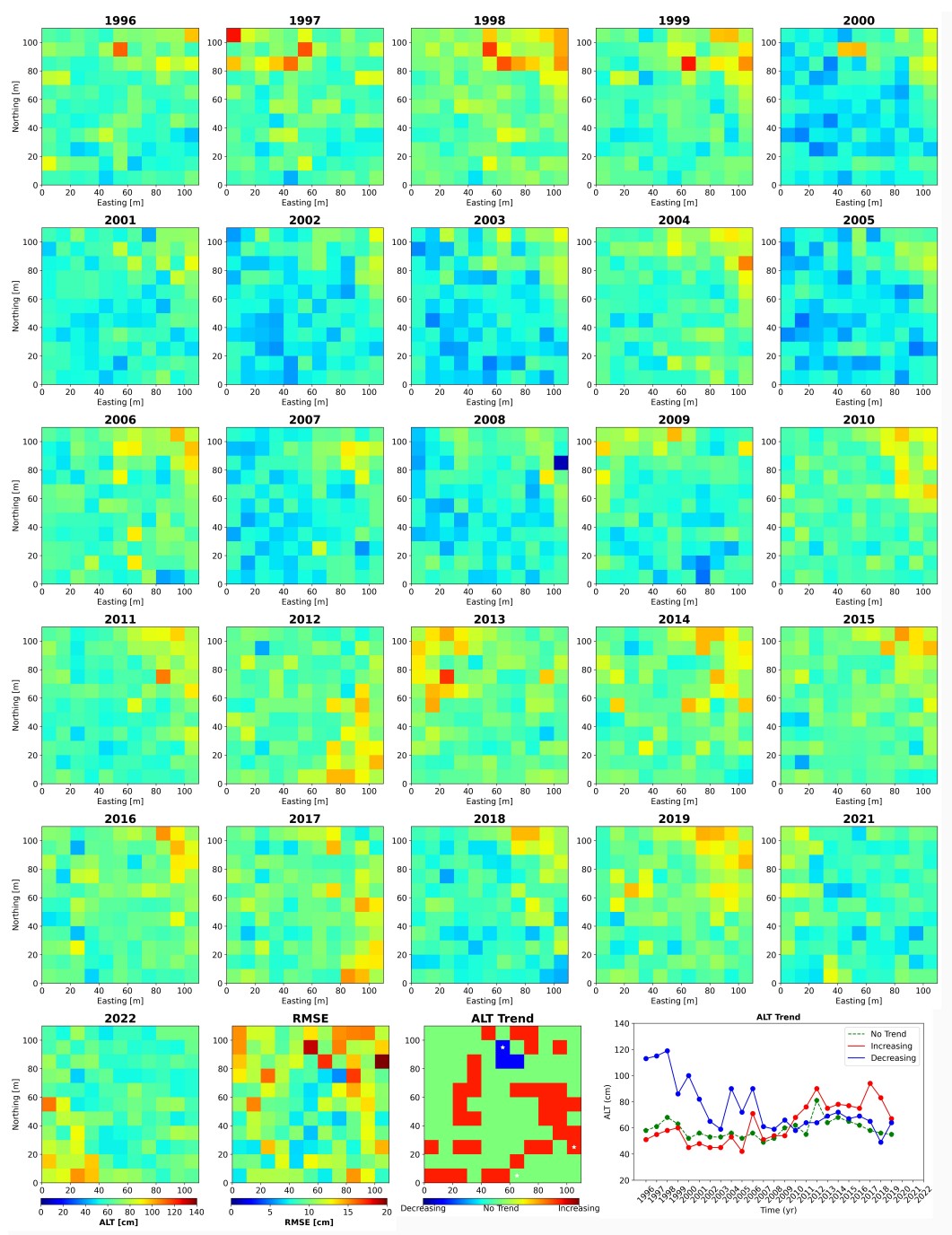

**Figure 7.** ALT variation of CALM site U8, RMSE of each cell relating to its annual average from 1996 to 2022, ALT trend and ALT time-series for three selected cell (10x10 m) shown by white star.

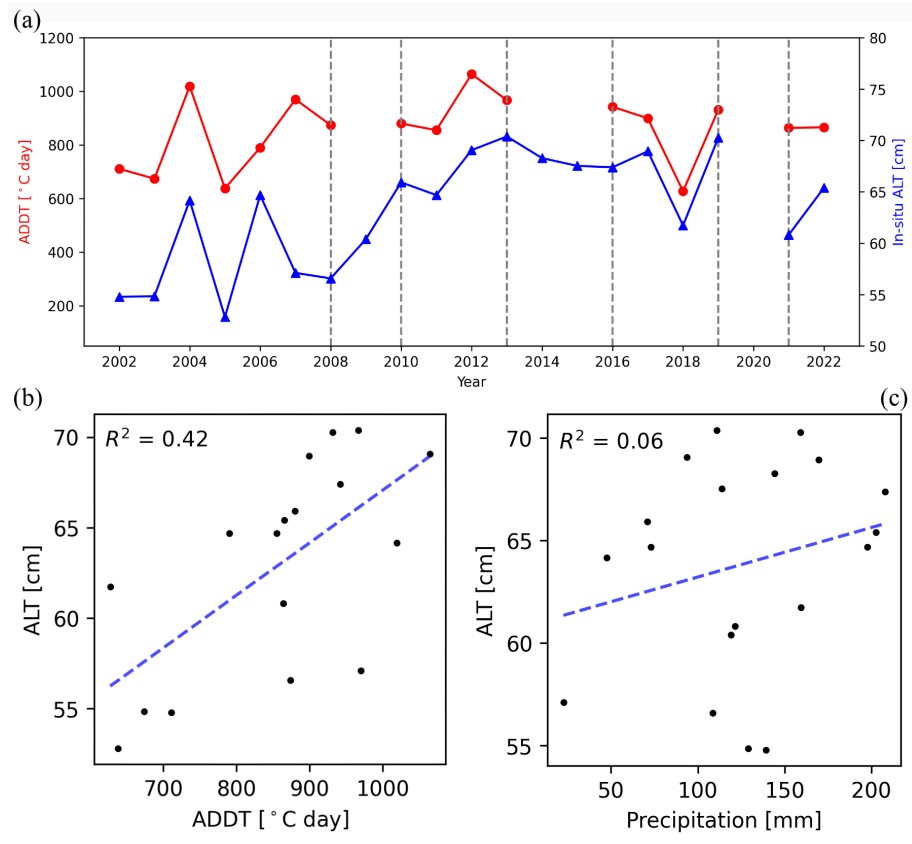

**Figure 8.** (a): Relation between ADDT and ALT from 2002 to 2022 in CALM site U8 and Sagwon station. Red circles show ADDT. Blue triangles show in-situ ALT. (b) scatter plot of ALT vs ADDT. (c) scatter plot of ALT vs precipitation. R-squared of relation is shown in top-left of panels. ADDT and precipitation are calculated from June 1 to September 1 of each year to be consistent with ALT measurements.

Alaska's discontinuous permafrost zone and has abundant black spruce, up to 5 m in height. The technique was not successful, as phase coherence was not maintained in successive SAR images, perhaps reflecting the relatively short wavelength (C-band) of the Sentinel-1 SAR instrument (see next section). Average spatial and temporal coherence maps for these two sites are compared in Figure S1.

### 5.6 Limitations and Future Research

Four aspects of our approach limit its utility and are an obvious focus for future research.

1. Decorrelation of InSAR phase is the main limitation of our technique. Accurate InSAR measurements require a high degree of coherence, a measure of the correlation in radar phase between the two SAR images. Decorrelation occurs due to temporal changes in surface scattering properties, changes in viewing angles, and noise in the SAR data (e.g., Schaefer et al., 2015). C-band InSAR has demonstrated its ability to monitor deformation over continuous permafrost region at higher lati-

tudes (see Previous Work, and this study). Wang et al. (2020) compared the efficiency of Sentinel-1 for monitoring permafrost deformation in discontinuous permafrost regions. They concluded that Sentinel-1 InSAR time-series performs effectively over permafrost landscapes mainly beyond the tree line, such as tundra, tundra wetlands, and less developed shrub-tundra areas. However, the outcomes and precision are less favorable in shrub-tundra and forest-tundra environments. Our results are essen-

tially the same: temporal and spatial coherence in our main study area, north of the tree line near CALM site U8 (almost entirely free of trees, (https://www2.gwu.edu/~calm/data/webforms/u8_f.html) are significantly better than those obtained in the discontinuous permafrost region near Fairbanks, Alaska (section 5.5). Significant decorrelation also occurred around CALM site U18 ($\sim$15 km southwest of Fairbanks, Alaska) during the 2023 thaw season. Land cover here is open black spruce forest (https://www2.gwu.edu/~calm/data/webforms/u18_f.htm). Longer wavelengths such as L-band may be more useful in densely

vegetated terrains. The NISAR mission, scheduled for launch in 2024, with its L-band wavelength and repeat frequency of 6-12 days, should prove useful for more densely vegetated discontinuous permafrost regions.

2. The spatial and temporal resolution of models that allow estimation of key ancillary parameters may limit accuracy in some regions, for example soil parameters from the GLDAS model and atmospheric parameters from ERA-5. The spatial resolution of GLDAS' soil parameter model is 0.25 degrees, an area that spans our study entire study area in the Alaska north

slope. The temporal resolution of ERA-5 is adequate, but its spatial resolution limits local analysis.

3. Accurate, dense and widespread porosity-depth profiles would improve ALT estimation from remotely sensed data. In particular, empirical and statistical models of soil properties calibrated with in-situ data could significantly improve radar-based ALT models (e.g., O'Connor et al., 2020; Bakian-Dogaheh et al., 2020, 2022, 2023).

4. Variations in soil ice content and non-linear thaw season subsidence time series need to be considered (Zwieback et al.,

2024).

## 6   Conclusions

We used Sentinel-1 interferometric SAR data from 2017 to 2022 around CALM site U8 in Northern Alaska to measure thaw season subsidence and estimate active layer thickness with a widely used physical model that exploits the volume difference between ice and water. Limited ICESat-2 LiDAR data are consistent with InSAR estimates of seasonal subsidence. We do

not attempt to estimate long-term (multi-year) elevation change. Instead we estimate ALT at the end of each thaw season and compare its yearly evolution, avoiding issues of decorrelation of the radar signal over the winter season.

ALT estimates in our study area range from $\sim$20 cm to more than 150 cm, similar to in-situ measurements at the CALM site and previous remotely sensed estimates. Agreement with the later part of the quarter century-long CALM time series is notable and suggests that annual ALT estimates from satellite InSAR can be effective at monitoring longer-term permafrost

health, at least for Alaska's continuous permafrost zone north of the tree line. However, the technique was not effective in the discontinuous permafrost region of central Alaska near Fairbanks, reflecting decorrelation of the C-band radar signal, probably from heavy tree cover. At the northern study site, ALT shows high spatial and temporal variability in both the satellite and in-situ data sets, sometimes changing dramatically between adjacent 10 m cells. Subsidence rate also varies significantly between

closely spaced points, ranging from ∼2-18 mm/month at our northern study site during thaw season. The reasons for such

high spatial and temporal variability of ALT are not clear and warrant further research.

*Code and data availability.*

Meteorological data from NOAA climate data online tool (CDO) is publicly available at (https://www.ncei.noaa.gov/cdo-web). Copernicus GLO-30 Digital Elevation Model is publicly available through (https://portal.opentopography.org). Sentinel-1 data are publicly available through Alaska Satellite Facility (https://search.asf.alaska.edu/#/). Interferograms were formed using

Alaska Satellite Facility's Hybrid Pluggable Processing Pipeline (https://hyp3-docs.asf.alaska.edu/using/vertex/). Time-series analysis is done by using (https://github.com/insarlab/MintPy) in OpenScienceLab JupyterHub computing environment (https://opensciencelab.asf.alaska.edu). ERA-5 data for tropospheric corrections are available at (https://cds.climate.copernicus.eu). Soil fraction data are available at (https://ldas.gsfc.nasa.gov/gldas/soils). In-situ active layer thickness data from CALM sites is publicly available at (https://www2.gwu.edu/~calm/). The python code for ALT estimation is archived at (Sadeghi Chorsi,

2023).

*Author contributions.*

TSC, FJM, and THD conceptualized the overall study. TSC performed the data processing and analysis. TSC, FJM, and THD wrote and edited the manuscript. THD provided financial support for the study.

*Competing interests.*

The contact author has declared that none of the authors has any competing interests.

*Acknowledgements.* This project was funded by grants to THD from the NASA Earth Science program (grant #80NSSC22K1106) and the Army Corps of Engineers (USACE Engineer Research and Development Center, Cooperative Agreement W9132V-22-2-0001). We thank Irena Hajnsek, Malte Vöge, Roger Michaelides, and Vincent Boulanger-Martel for detailed reviews that greatly improved the manuscript. We appreciate Regula Frauenfelder for editing the paper.

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
