# Peer review of "Toward Long-Term Monitoring of Regional Permafrost Thaw with Satellite InSAR"

_EGUsphere, 2023_

## Author Response (AR2)

**Response to Editor**

We appreciate the editor for thorough feedback and recommendations. We have incorporated suggestions and technical corrections, which we believe have enhanced the manuscript. Below we give a detailed response addressing each point raised.

**Comment**: Dear authors, thank you for your point-to-point update of your manuscript and your according responses in your Author response. I am mostly satisfied with your updates. However, I need some clarifications of the points below, before I can accept your paper for publication.

**Answer**: Thank you for your valuable feedback and recommendations. We have implemented your comments and provided detailed responses to each point below.

**a)** While I see your answers to the first specific point by Referee no. 2 (Ms Hajnsek) concerning influence of soil moisture and precipitation, it is not clear to me how you have addressed these points in your manuscript. Please upload a clarification, e.g. pointing out where and how you have incorporated your reply in the text.

**Answer**: We agree with the editor and reviewer and we addressed this comment in lines 82-89 explicitly. We noted in the "Methods" section (lines 82-89) that variations in soil moisture could also decorrelate the radar wave's coherence. To mitigate this possibility, we looked for noisy data, which is partly due to such moisture changes, and excluded these from our analysis. We employed two criteria to assess noise. First, we manually reviewed interferograms and selected those that were not noisy (lines 85-87). Second, we assessed the spatial and temporal coherence of the chosen interferograms to ensure they all had high coherence (lines 85-87). This information is detailed in the supplementary materials. Additionally, we note that data were collected in a period most likely to retain soil moisture: immediately after snowmelt, when the soil column is saturated, and near the end of thaw season, when

the soil column is likely still to be wet due to the complete thawing of any residual ice in the active layer. Consequently, we believe the soil column is likely to be saturated during our measurement period, hence it is unlikely that soil moisture changes affected the results (lines 86-90). Two other points are relevant. First, we are confident that our measurements accurately reflect real elevation changes, supported by agreement with ICESat-2 height measurements. Second, we are confident that our models accurately reflect true active layer thickness, supported by consistency with independent ground-based ALT measurements.

**b)** The same applies to her comment concerning "the estimation of ALT". Please upload a clarification, e.g. pointing out where and how you have incorporated your reply in the text.

**Answer**: We agree with the editor and reviewer and we addressed this comment in lines 285-287 and 319-322 explicitly. We selected the model parameters based on available in-situ data and published literature. We then estimated ALT and compared it to in-situ ALT measurements. Lines 291-321 were added to demonstrate the agreement between the estimated ALT and the in-situ data. This agreement is depicted in Figure 6. In Figure 8, we explored the relationship between accumulated degree days of thawing (ADDT) and precipitation with _in-situ_ ALT. However, this relationship was found to be weak. Therefore, given that our estimated ALT aligns well with in-situ ALT (Figure 6) and that the long-term in-situ ALT measurements (2002-2022) show no correlation with ADDT and precipitation (Figure 8), we suggest that other factors might be influencing the results (lines 320-322).

**c)** I acknowledge your attempt to shorten section 2 "Pervious work" according to a reviewer comment. However, in doing so, you seem to have overdone it a bit. You do see that you have gathered the most relevant references in a table, which is nice. However, I need you to add a better framing of the information                 given                 in                 the                 table.

**Answer**: We agree with the editor and we extended the "previous work" section to address this challenge (lines 38-45).

We think the reviewer was correct to point out that a long summary of previously published work is inappropriate in a journal whose primary focus is new scientific results. The challenge here is that there are now nearly three decades of work exploiting satellite sensing of permafrost. While we build on much of this work, and feel it is useful to cite all relevant prior published work, we also have something new to contribute, hopefully better described in this version of the paper.

In reviewing these other published papers, none of them are as thorough as we are at referencing prior work. We believe we have done a service to the community at compiling the references to all these studies in one widely read journal, even if it is just in table format. At the same time, Cryosphere is not a review journal, so we feel it is not appropriate to use up text to summarize these papers. Readers of Cryosphere can easily to that themselves with the information summarized in our Table.

Our compromise is to specifically reference the four or five papers most relevant to our study, where InSAR is used to measure seasonal elevation changes, then use a physics-based model to estimate ALT. In the later sections of our paper, we reference specific aspects of these papers where appropriate, either where use the same formulations, or differed from these approaches.
* * *
**1. Response to Dr. Malte Vöge, referee 1:**

*We appreciate the reviewer for their thorough feedback and recommendations. We have incorporated the majority of their suggestions, which we believe have enhanced the manuscript. Below we give a detailed response addressing each point raised.*

**General Comments:**

The paper describes a method to estimate active layer thickness in permafrost regions based on subsidence rates in the thaw season measured by the means of SAR interferometry. A test of the method

is presented for a site in Alaska and the results are compared to ICESat-2 measurements (although only for one of 6 seasons) and in-situ data.

The paper is well structured and written in good and comprehensible English. The introduction provides the relevant background, although the "Previous work" section is rather lengthy and could focus more on the most relevant publications. The methodology section provides a thorough explanation of the applied method and the data that has been used. The discussion of the result from applying the method to the test dataset is thorough. It addresses relevant aspects of the comparison with the ICESat-2 data and the in-situ data and provides error estimates for the derived ALT values.

The paper contains a separate section for limitations of the applied technique and future research that should be conducted. This includes both aspects of InSAR measurements in permafrost regions as well as the ALT estimation from displacement time series. The conclusion is rather short, however, it summarizes the most important findings and limitations and future plans/work have already been discussed in the previous section.
* * *
**Specific Comments:**

**Comment:** The used InSAR processing chain is briefly explained with extra sections for reference point selection and atmospheric delay correction. While I see the need to discuss the reference point selection considering the specific challenges in permafrost regions, I do not see any specific challenges in terms of atmospheric delay correction. Therefore and as this is a standard part of InSAR time-series processing, this part might be shortened and could be moved to the first subsection of the "InSAR Data Processing" section. Instead, a discussion of the (for InSAR time-series processing) rather short time-periods per season could be added, which implicates the expected accuracy of the measurements.

**Answer:** While we understand the motivation of this comment, after some deliberation, we decided to only apply minor modifications to this section. This decision was made for the following reasons: his section was added after the reference point selection section because we wanted to show the steps of time-series processing. Moreover, finding a suitable reference point for all six years was difficult, and we had to select  a point on top of an outcrop whose elevation was ~100 m higher than river plain. Stratified atmospheric delay was present in a number of interferograms and the height difference between the reference point and the study site became an issue. Thus, applying atmospheric corrections was necessary. Applying two methods (ERA-5 and GACOS) to correct this effect, we preferred to use ERA-5 since GACOS added more noise into interferograms. We think that the better performance of ERA-5 could be due to its higher temporal resolution (1 h) regarding to GACOS (6 h). In other words, while most of the tropospheric correction was standard, some aspects were unique to this study, and we feel that this is an important part of the study. This is mentioned in lines 103-105.
* * *
**Comment:** The ALT model is presented in an understandable manner. The equations support the descriptions and allow the read to follow the derivation of the model. Relevant citations are given for previous work that the derivations are based on. What this section is missing, though, is to point out what these derivations add to the methods applied in the cited publications. Most of the derivation appears to be taken from Liu et al 2012, who also derived ALT from InSAR. Although Liu et al 2012 is cited several times, it should be made more clear that the applied approach largely follows the method presented in this paper, and as mentioned above, it should be made clearer, which parts of the method have been added by the authors.

**Answer:** This comment is now addressed in the revised version of the paper in lines 143-146 and 163-166. We explicitly mentioned in the paper that we followed the virtually identical procedure described in Liu et al. (2012) in sections 4.3.1 to 4.3.4. We do not consider secular (long-term)

displacement signals in model because we analyze the thaw seasons of 2017 to 2022 separately. This is the major difference between our approach and those described in Liu et al. (2012), Schaefer et al. (2015) and where seasonal and interannual trends were estimated simultaneously (lines 163-166). This methodology helps us to compare yearly InSAR-derived model with yearly in-situ data. Assessing the effectiveness of technique by comparing with two independent data sets; available ground truth and ICESat-2 ATL08 product are two other contributions that are added in this paper. We mentioned these explicitly in lines 143 to 146 and 166-166.
* * *
**Comment:** In the "Limitations and Future Research", points 2-4 could be more elaborate. Especially for point "4." about porosity-depth and water content models based on in-situ data, it could be extended with more details on how this could be done and what data could be available for this in the future.

**Answer:** This comment is now addressed in the revised version of the paper. We added additional explanations to better discuss this point. See section 5.6 (lines 336- 361).
* * *
**Technical corrections:**

**Comment:** Line 194: "To compare with relative InSAR data, we subtracted the two available ICESat-2 height data and assign the first date's height as zero elevation. This is a reasonable assumption because the two datasets have comparable start dates, June 8 for ICESat-2, and June 3 for                                                                                                                           SAR."

This is not quite clear. Do you simply mean that the ICESat-2 height difference is compared to the InSAR displacement, given that the temporal gaps for both datasets in this comparison are similar? Please clarify this!

**Answer:** Yes. The temporal resolution of ICESat-2 data is 91 days for repeat tracks. Here, we subtract the elevation of two area (50m x 50m) in two days which are 91 days apart and are close to Sentinel-1 acquisition times. We put height difference in the plots of InSAR time-series and compare together. We basically tried to find locations that have both LiDAR and Radar data and considered some constrains regarding to ICESat-2's uncertainty. This is now discussed more in the section on ICESat-2 data types, processing and results. It is clarified in lines 126-133 and 230-235.
* * *
**Comment:** Line 281, "Applying atmospheric corrections to C-band radar images improves signal to                                                                                                   noise                                                                                                     ratio.":

While this is certainly true, this has not been shown in this paper, as no comparison of InSAR results with and without atmospheric corrections are presented. While this could be mentioned in the methodology section, this sentence should be removed from the conclusion!

**Answer:** This comment is now addressed in the revised version of the paper. We put the sentence in "atmospheric delay correction" section.

**2. Response to Dr. Irena Hajnsek, referee 2:**

We thank the reviewer for the detailed comments and suggestions. We implemented most of them and we think this improved the manuscript. Below is a point-by-point response.

General Comments:

The main objective of this manuscript is the satellite monitoring of deformation over permafrost regions to relate the deformation to the change in active layer thickness (ALT). The remote sensing method is based on SAR interferometry using Sentinel 1 data from 2017 to 2022 and shows surface

decrease in cm continuously over the years. However, the relation to the accumulated degree days of thawing (ADDT) is statistically not significant and the authors point out some critical issues that need to be considered when performing the relation.

In general, the paper is well written and follows a logic structure. The study makes perfectly sense to use SAR interferometry to map large scale deformations over the vast permafrost areas and I support the publication of such a study. I have several suggestions to improve paper by considering some specific points:

Comment: SAR interferometry at C-band makes only sense over regions with low vegetation – as it has been already stated by the authors and would even make more sense if longer wavelength would be available to have a lower sensitivity to vegetation cover. However, my concern is here also that also a change of soil moisture can strongly impact the coherence amplitude as well as the phase. How do the authors make sure, that soil moisture changes over months would not affect the deformation measurements?

Answer: We agree with the reviewer that soil moisture is an important parameter to consider and we mentioned it in lines 64-69 in "Methods" section. In the future this can be addressed explicitly with multi-wavelength observations, where soil moisture and penetration depth can be modeled. The upcoming NASA-ISRO NISAR mission will help in this regard. For now, we are reasonably confident that our measurements and model results do reflect real elevation change, based in part on their agreement with two independent data sources, ground-based measurements of ALT, and ICESat-2 height measurements.

In addition, our data selection may reduce the impact of soil moisture. The wavelength of the Sentinel C-band SAR is short enough that penetration of the upper soil horizon is likely minimal unless the soil is extremely dry. We chose SAR data collected shortly after start of thaw season, when soil conditions are almost certainly saturated. For example, in 2017, the first Sentinel-1 SAR

data used in our time series was acquired June 12, while thaw began here on May 24 according to the Sagwon meteorological site. As the study area is an ice-rich continuous permafrost region, it is likely that soil remained wet through most of the thaw season as sub-surface ice continued to melt, but we cannot preclude locally dry conditions.

We also agree with the reviewer on the importance of precipitation. During the 2017 thaw season, the average precipitation was ~2 mm/day from June 12 (the beginning of our time series) until September 4 (the end of the time series). ~37% of the data in this period experienced precipitation. However, we could not assess the study area in detail in terms of precipitation because of the coarse spatial resolution of available weather models (e.g., ECMWF).
* * *
Comment: As a reference ICESat-2 data where used for a particular time – as the data have been not available over a longer period. In my opinion this makes sense, as they have highly precise height measurements. However, the height variance should be evaluated, as it lies in the range of the deformation measured.

Answer: This is an important point and is addressed in the revised version of the paper under "Validation of InSAR Surface Displacement Estimates with ICESat-2 Data" section in lines 243 to 252. We also discuss ICESat-2 uncertainties, including a comparison of the ATL06 ('land ice height') and ATL08 ('terrain and vegetation height') data products in supplement material. The precision and accuracy of ICESat-2 and specifically the ATL08 data product are now described in detail. Neuenschwander et al. (2019) outline various sources contributing to the uncertainty of the ATL08 height product. These include the precision of the instrument's ranging capabilities, uncertainty in radial orbital positioning, knowledge of geolocation, atmospheric forward scattering, uncertainty in tropospheric path delay, local topography, sampling error, background noise, vegetation, and misidentified photons. The precision of the instrument's ranging primarily depends

on the width of the laser pulse and uncertainties in timing electronics. The reported uncertainty is based on error propagation of each of the listed error sources. In areas where the surface is relatively flat, with slopes of less than 1 degree, the effect of pointing error on the elevation measurement is minimized and is less than 25 cm, taking into account a radial orbital uncertainty of 4 cm and tropospheric path delay uncertainty of 3 cm. The ATL08 height product also provides information on the standard deviation of multiple terrain points for within a given 100-meter segment. The reported standard deviation for four chosen points in segments in our study areas ranges from 0.11 m to 0.24 m. This parameter mainly reflects surface roughness. Finally, the Circumpolar Arctic Vegetation Map (CAVM) classifies vegetation in terms of elevation. Part of the standard deviation in a given height estimate is due to the varying photon returns from the bottom, middle and top of the vegetation. In our study area, this source of scatter is minimal, since there is no tree cover. In summary, our study area, with its minimal vegetation cover and flat terrain, is an ideal place to use the ICESat-2 ATL08 data product. Moreover, it is completely independent of our Sentinel InSAR-based elevation change estimates.
* * *
Comment: For the estimation of ALT some model assumption has been used, that where justified by references that have been published. In my opinion also this makes sense, however as statistically no relation was found, it would have been useful to adjust some model parameter assumptions to retrieve a better fit and to analyses these results.

Answer: There are many parameters that affect the ALT remote estimation model. These include organic matter mass, organic matter porosity, organic matter density, sand fraction of soil and resulting mineral porosity, soil moisture, and maximum root depth. Each of these parameters have different impacts on ALT separately but also have interactions with each other. We conducted a simple statistical study to analyze the impact of some key features that could be quantified (e.g.,

precipitation, temperature) on in-situ ALT. A more comprehensive study is warranted but is beyond the scope of this study, however we hope to study this in future research.
* * *
Specific Comments:

Page 1, Line 1: Please specify the investigation area – replace 'part' with the specific area

Answer: This comment is addressed in the revised version of the paper, it is changed to "specific area" in line 1.
* * *
Page 1, Line 13: The sentence …'ALT is expected….is important' do not give the reasoning why remote sensing should be used to monitor permafrost features. Please rephrase.

Answer: This comment is addressed in revised version of the paper. We rephrased the sentence to emphasize the importance of the remote sensing (lines 16-20).
* * *
Page 1, Line 16: Interferometry is also used for glacier velocity estimation, which is an important application. Please add.

Answer: This comment is addressed in the revised version of the paper. We added "glacier velocity estimation" as an another InSAR application and give an example in line 27.
* * *
Page 2 under previous work: Please also add Zwieback et al. https://tc.copernicus.org/articles/12/549/2018/ as an example of using double differential SAR interferometry for height difference estimation and Bernhard et al. doi: 10.1109/JSTARS.2020.3000648 also as an example to use height difference for RTS estimation.

Answer: This comment is addressed in the revised version of the paper. Page 3, table-1 now shows the previous work which includes two mentioned paper.
* * *
Page 3, Figure 1: Please also explain what positive values in Figure 1b means.

Answer: This comment is addressed in the revised version of the paper under caption of figure 1.
* * *
Page 4, Line 68: Please explain if 12 or 6 day repeat pass scene where selected.

Answer: This comment is addressed in the revised version of the paper. Temporal resolution of the data is 12-days in a descending geometry (lines 67-69).
* * *
Page 4, Line 70: Coherence decorrelation can also occur during change in soil moisture and is then decorrelating during the thaw season.

Answer: This comment is addressed in the revised version of the paper (lines 64-67).
* * *
Page 4, Line 76: Deformation only in LOS measured with the method. Could this be also a reason why the correlation with ADDT is not so high?

Answer: The ascending geometry data was not available for the study area. So, we were not able to solve deformation for horizontal and vertical components simultaneously. However, since the area is tectonically stable, we can assume the observed deformation sensed is dominated by vertical component displacement. We convert descending LOS to vertical component by considering local incidence angle. We think the moderate correlation between ALT (both in-situ and remote) and ADDT is reasonable since there are other physical parameters that affect ALT. Just note that the in-situ data had moderate correlation with calculated ADDT from station ~30 km south of the CALM site (See Figure 8).
* * *
Page 4, Line 79: The word 'double-difference' interferometry describes the subtraction of two interferometric pairs and not a single interferometric acquisition. Please correct.

Answer: This comment is addressed in the revised version of the paper. It is now corrected on page 5, line 82.
* * *
Page 5, Line 103: How good is the atmospheric corrections?

Answer: We observed some spatially-correlated tropospheric effects before applying corrections partly because of the location of the reference point. The reference point is relatively higher than most of the area located on top of the outcrop. The CALM site is near a river channel whose elevation is more than 100 m lower than reference point. Stratified atmospheric delay was present in a number of interferograms and the height difference between the reference point and the study site became an issue. Thus, applying atmospheric corrections was necessary. Applying two methods (ERA-5 and GACOS) to correct this effect, we preferred to use ERA-5 since GACOS added more noise into interferograms. We think that the better performance of ERA-5 could be due to its higher temporal resolution (1 h) regarding to GACOS (6 h). In other words, while most of the tropospheric correction was standard, some aspects were unique to this study, and we feel that this is an important part of the study. We rephrased section 4.1.3 to address part of this comment.
* * *
Page 5, Line 112: Throughout the manuscript the 'Figure-S1' appears. I could not find this figure! Please correct.

Answer: The figure-S1 (Now Figure S3 in revised paper) shows ICESat-2 ATL08 difference elevation for two repeat tracks and was put in the supplement material.
* * *
Page 5, Line 114: The question appears to me what height accuracy is expected with ICESAT-2? Could you please add the estimates?

Answer: This comment is addressed in the revised version of the paper by adding the estimates of uncertainty of ICESat-2 and standard deviation of ATL08 product in a separate table supplement material (Table S2 and S3). See also the note above.
* * *
Page 5, Line 116: please do not use abbreviation in the title.

Answer: This comment is addressed in the revised version of the paper.
* * *
Page 5, Line 118: As already stated the motion is not only appearing due to thawing in general but can also appear due to soil moisture changes, as the electromagnetic wave when the soil is dry is penetrating deeper in to the soil surface. The variation lies in the cm range, the same range as also the deformation appears. Please explain.

Answer: This is addressed above, see Answer to the first comment.
* * *
Page 6, Line 127: please do not use abbreviation in the title.

Answer: This comment is addressed in the revised version of the paper. We avoid using abbreviation in section title.
* * *
Page 6, Line 145: Table S1 does not exist. Please correct.

Answer: The Table S1 shows information of used interferograms in this study and was put in the supplement material.
* * *
Page 6, Line 149: What happens if you have vegetation on top of the model? Even tough the vegetation cover is short in height. This is not considered in the model.

Answer: During our study, which included tests in other areas, we observed high decorrelation due to vegetation in forested areas of central Alaska's discontinuous permafrost zone. Decorrelation was

low in our studied area, northern Alaska's continuous permafrost zone. In section 5.5 "Applicability to other Regions", we explore the efficiency of this method on vegetated areas in central Alaska (lines 329 - 335).
* * *
Page 7, Line 175: I am not sure but I assume that the letter roh is not correct and should be replaced by P.

Answer: The equation shows relation between organic density and depth and it is "rho".
* * *
Page 9, Line 192: Please correct Figure S-1 – I cannot find this figure in the manuscript.

Answer: The figure-S1 (Now changed to Figure S3) shows ICESat-2 ATL08 difference elevation for two repeat tracks and was put in the supplement material.
* * *
Page 10, Figure 3: It would be useful to have the std variation of the measurements also added into the plot. Another point that would be important to clarify how can the authors be sure that the change is not due to soil moisture change during the months.

Answer: This comment is addressed in the revised paper. We discuss on std of ICESat-2 products and put RMSE of linear fit on top left of the panels.
* * *
Page 14, Table 1: Units are missing.

Answer: This has been corrected in the revised version of the paper. We omit the table 1 in revised version of the paper because it has same information as figure 6. Also, the errorbars are added into figure 6 per request.
* * *
Page 15, Line 263: Please correct the Figure-S2!

Answer: The figure-S2 (Now named to Figure S1) show the coherence maps for two compared sites, Beta (located in discontinuous permafrost zone) and U8 (located in continuous permafrost zone) and was put in the supplement material.

**3. Response to Dr. Roger Michaelides:**

*We value the reviewer's detailed feedback and recommendations. We have integrated most of their suggestions into the manuscript, which we think have greatly improved its quality. Below, we provide a thorough response addressing each point raised by the reviewer.*

**General Comments:**

In this manuscript, the authors use Sentinel-1 InSAR data collected between 2017 to 2022 to derive estimates of seasonal subsidence over an area of continuous permafrost extent on the North Slope of Alaska with an SBAS time series inversion and using the approach detailed in Liu et al. [2012]. They then estimate the active layer thickness from these estimates of seasonal subsidence following the ReSALT model described in Liu et al. [2012] and compare them to in situ measurements and ALT estimates from the NASA ABoVE airborne campaign. They additionally use ICESat-2 data to validate InSAR displacement time series, finding agreement between the two measurement approaches.

In general, this manuscript is well-written and most methods and results are clearly described. While I support publication of this manuscript in this journal, there are several specific points that should be addressed before the manuscript can be considered for publication. Several major points numbered below with some associated comments, and minor comments are included in a separate section below.

**Specific Comments:**

> **Comment:** The manuscript would be strengthened with a more explicit explanation and description of the novel contributions made by this manuscript. In general, the methods employed are meticulously described, but these seem to mostly be previously published methods, and reference is made to other studies that investigated InSAR times series analysis and ALT estimation over the same study site (or broader North Slope region). Instead, I would recommend to the authors to emphasize the novel contributions of this study that either build

upon and extend the prior studies referenced here, or the novel insights made here that haven't been discussed elsewhere. For example, are there any changes made to the seasonal subsidence estimation approach or ReSALT model employed here as compared to the original Liu et al. [2012] paper? Or is the approach of characterizing total seasonal subsidence here substantively different from comparable approaches in Rouyet et al. [2019] (https://doi.org/10.1016/j.rse.2019.111236), Rouyet et al. [2021] (https://doi.org/10.1029/2021JF006175) or Michaelides et al. [2021] (https://doi.org/10.1029/2020EA001538)? By emphasizing these differences, the authors will be able to emphasize the novelty of this study more effectively.

**Response #1:** We appreciate this comment and took it to heart. We have made several modifications in the revised paper to address this concern. These chances can be found in Sections 4.3.2, 5.2, and 5.3. These changes are briefly summarized here: There are three main differences with previously published work. First, changes were applied to the ReSALT model, in that we do not consider interannual (long-term) changes in the active layer thickness (ALT). The main reason is that for InSAR, connecting the phase measurements across the winter season is challenging. We have taken a more conservative approach and just consider changes within a thaw season. We show that this is still a useful way to monitor long-term permafrost health, since by definition long-term loss of permafrost means increased ALT. Second, we establish that height changes measured by the InSAR SBAS method can be usefully compared with the ALT08 product of ICESat-2 over relatively flat terrain. While the nominal uncertainty of the ALT-08 product is of the order of (or worse than) the changes of interest, many of the error sources are systematic and hence cancel or are greatly reduced by a simple differencing procedure. To our knowledge, this is the first direct comparison of the ATL-08 product with InSAR-based techniques for permafrost. Third, we do a long term (nearly three decade)

comparison of our results to ground-based measurements, confirming the accuracy of our approach and suggesting that at least at this location, there are no significant long-term trends in ALT.
* * *
**Comment:** Section 4.3: As best as I can tell, Section 4.3 is more or less a condensed discussion of the methodology presented in Liu et al. [2012]; the only substantive difference that I can find is Equation 4, which is expanded in Liu et al. [2012] to simultaneously solve for the linear deformation trend as well (more on this in a separate comment). It is probably sufficient to just state 'we followed the identical procedure described in Liu et al. [2012]" and mention this small change, as the rest of the detail in this section does not differ from Liu et al. [2012].

**Response #2:** This comment is now addressed in the revised version of the paper. We mentioned this in lines 143-146.
* * *
**Comment:** As a stylistic comment, there is a lot of detail devoted to describing prior work or describing the methods employed in this manuscript (which were developed in prior studies and thus do not require as much detail here), and comparatively little detail devoted to describing the results of this study, discussing their significance, limitations in the study, or potential avenues of future work. It might strengthen the manuscript if this relative imbalance in detail between sections were lessened.

**Response #3:** This comment is now addressed in the revised version of the paper. We shortened the previous work section by putting most of the information into table (Page 3). We added more information on methodology describing the ICESat-2 ATL08 product, and described the study area in

more detail. We also discussed (figure 7) the in-situ ALT data for the U8 site from 1996-2022 to explain spatial and temporal variation of ALT (see below, response # 8).
* * *
More analysis of the retrieved parameters in the Results section would, in my opinion, strengthen the manuscript and potentially help shed important insights into the relative limitations of various InSAR-based methods for retrieving permafrost physical properties. There are several concrete examples that I can think of that are enumerated below:

**Comment:** More detail would be appreciated on the use of ICESat-2 to evaluate InSAR results. Additionally, it would be worth reporting the standard deviation and/or measurement uncertainty for the ATL-08 data set, as this is likely comparable to, if not larger than, the inferred deformation of 2-6 cm. It would be helpful to compare this to the Michaelides, Bryant, et al. [2021] study (**https://doi.org/10.1029/2020EA001538**) which compared ICESat-2 and Sentinel-1 over the North Slope of Alaska.

**Response #4:** This comment is addressed in the revised version of the paper. We report the uncertainty and standard deviation of the ATL08 product, and compare to the Michaelides et al. (2021) study. In section 5.2 (lines 243-252), we discussed on different ICESat-2 product and studies alongside InSAR. We also comapre between ATL08 and ATL06 in study area and put the results in supplement material.
* * *
**Comment:** Reporting linear rates in Figure 3 and section 5.2: Reporting rates as high as 20 cm/year is somewhat misleading, in that we do not expect this linear deformation rate to persist throughout the year (i.e., this is an episodic deformation that persists for only a few months out of the year). This point should be acknowledged in the text so that readers do not misinterpret this.

**Response #5:** This comment is addressed in the revised version of the paper. We clearly stated that the rate of linear deformation is not anticipated to persist throughout the entire year, as it occurs episodically during the thaw season. We reported the rates in mm/month to avoid confusion. See caption on figure 3.
* * *
**Comment:** In addition to the above point, it is also worth noting that the linear rates reported here are different from the interannual trends reported in Liu et al. 2012, Schaefer et al. 2015, and Michaelides et al. 2019. These latter trends are estimated by modifying equation 4 in this manuscript to simultaneously solve for the interannual trend, rather than estimating the interannual rate by fitting a linear regression to the SBAS results (which is what I assume is being done in section 5.2, although it isn't clear).

**Response #6:** This comment is addressed in the revised version of the paper and is summarized in Response #1, above. Note that we do not estimate interannual rates. Figure 7 in the paper does compare ALT at the test site over nearly three decades, but no significant long-term trend is apparent in these data.
* * *
Figure 5: How do the seasonal amplitude results from equation 4 compare to the total range of deformation from the SBAS results (figure 3)? It would be illustrative to compare these results to verify whether both methods seem to be capturing the same deformational processes, or if not, whether one approach (e.g., SBAS vs. ADDT model) is more favorable, or systematically underestimating/overestimating compared to the other method.

**Response 7:** We plot SBAS vs ADDT model derived from Stefan equation for U8 site. Both models agree considering standard deviation, but ADDT model generally overestimated the deformation.
* * *
**Comment:** Figure 5, continued: The large interannual variability in inferred seasonal subsidence (and hence ALT) is somewhat concerning; while ALT can considerably vary spatially at the scale reported here, I would not expect ALT to vary this considerably from year to year for a particular location. We would expect a stronger correlation from year to year for ALT values, as ALT tends to correlate with physiographic and environmental condition (e.g., slope, aspect, vegetation cover/type, soil saturation, proximity to wetlands) most of which should not vary considerably from year to year. I suspect that there is more going on here in the data; perhaps due to the use of decorrelated interferograms or interferograms with unwrapping errors. For example, for ALT 2020 in the bottom left corner, the regions of ALT >100 surrounded by ALT<25 seems somewhat implausible, but when we consider that these areas are in an area of disconnected wetlands, this may reflect a phase unwrapping artifact. ALT 2021 and 2022 seem the most plausible; how many scenes for each year were used in the SBAS inversion?

**Response #8:** We agree with the reviewer in the sense that one would not *expect* to see significant year-to-year variation in ALT at one location (that was our initial expectation). But in fact the available data show the opposite, and it cannot be a only due to phase-unwrapping artifact: inspection of ground-based ALT measurements at the U-8 CALM site show a surprising amount of year-to-year variation. This is now shown in Figure 7 and discussed in the manuscript.
* * *
**Comment:** Figure 6: Since uncertainties for ALT are calculated for this study (and included in the UAVSAR dataset of Chen et al. [2022]), they should also be included as error bars on figure 6 to visualize the degree to which the mean ALT estimates from each dataset diverge from each other (or

are within the uncertainty). Further, to intercompare retrieved ALT from this study and Chen et al. [2022], it would be more accurate to apply the same spatial averaging schema to both datasets in determining the mean ALT over a 100 m radius circle, rather than doing this for the 'Chorsi dataset' but not the 'Chen dataset'.

**Response #9:** This comment is now addressed in the revised version of the paper. We clarified in lines 270-273.
* * *
Several more minor comments are listed below:

**Comment:** Section 2 Previous work: It is appreciated that a thorough introduction to previous work using InSAR and other geodetic measurements to characterize permafrost is given here. However, the section reads a little repetitively, as each sentence is in the form 'Author X used … to …" Perhaps this section could be condensed by keeping all current citations but grouping together similar studies into thematic topics, such as which studies emphasized different technical applications, or had a particular scientific focus (e.g., fire or soil moisture).

**Response #10:** We address this comment in the revised version of the paper by condensing and putting most of this information in a table. Now table-1 covers all previous work material in different categories including scientific focus, technical applications, used dataset and study area.
* * *
**Comment:** Typo: Second sentence in Study area section: 'Brookes range' should be 'Brooks Range'.

**Response # 11:** This typo has been fixed, thanks for catching it (line 44).
* * *
**Comment:** 23 cm organic layer thickness: Where does this estimate come from? Over what spatial scale is this assumed representative? OLT can vary considerably over the scale of a few meters.

**Response #12:** The organic layer thickness estimate came from metadata at CALM site used in this study, cited in the manuscript: https://www2.gwu.edu/~calm/data/webforms/u8_f.htm. We assume that the whole area has the same organic layer thickness. We agree that OLT can vary significantly, but unfortunately, better stratigraphic data are not available. This is one of the main limitations of the technique. In addition to organic layer thickness, improved data on porosity and bulk density would be useful. These challenges are now discussed in the limitation sections of the paper. We referred to CALM site meta data in line 54 under section 3 (Study area).
* * *
**Comment:** Mention of APEX site seems a little disjoint and unrelated to the results presented; since there is no analysis for this site in the manuscript, I would suggest to remove this.

**Response # 13:** We applied the identical technique to the APEX site, in the discontinuous permafrost zone of mid-Alaska, but did not get good results because the C-band radar decorrelated in this forested region. We appreciate your suggestion but other reviewers have asked whether the technique we describe will work over broader regions. To accommodate all reviewer comments we have placed this section in the supplement. We put this information in separate section (5.5 Applicability to other Regions) to address all reviewers comments.
* * *
**Comment:** Figure 1: Why is red box square but panel b rectangular?

**Response # 14:** The red box is now shown in both 1a and 1b to clarify.
* * *
**Comment:** Section 4.2: This is the first time ICESat-2 is mentioned after the abstract; it would be good to mention how ICESat-2 will be used in this study in the introduction.

**Comment #15:** This comment is now addressed in the revised version of the paper, with information in mentioned in the introduction (lines 30-35).
* * *
**Comment:** Section 5.5: Perhaps the authors could expand upon points 2-4, as they are only 1-3 sentences. Additionally, most if not all of these discussion points made here have been made previously in other publications, which could be cited here. The discussion of the APEX site seems superfluous and can probably be removed from the manuscript. On the discussion of more accurate porosity and water content depth profiles, it would be worthwhile to cite the work by Bakian-Dogaheh and others on this exact topic in this same study region (e.g., https://doi.org/10.3334/ORNLDAAC/1759, https://doi.org/10.1088/1748-9326/ac4e37, https://doi.org/10.3334/ORNLDAAC/2149).

**Answer:** This comment is now addressed in the revised version of the paper. We expanded those points and referred to the above mentioned studies (See section 5.6). For the APEX site, see Response #13.

**4. Response to Dr. Vincent Boulanger-Martel:**

*We appreciate the reviewer's thorough feedback and suggestions. We have incorporated the majority of these suggestions into the manuscript, which we believe has enhanced its quality. Please find our detailed responses ("Answer") to each point below.*

**General Comments:**

The authors present an interesting study that aims to estimate the thickness of the active layer using InSAR analyses conducted over a 5-year period from 2017 to 2022. This study uses SBAS analyses to monitor seasonal ground deformations along the LOS. Deformations are the used to provide an estimation of year-to-year changes in the thickness of the active layer. An analytical solution is used to infer the thickness of the active layer from ground deformations and the annual thawing index observed at a nearby weather station. As expected, the maximum subsidence are observed at the end of the summer. The inferred thickness of the active layer is then compared to the thawing indices. Result show important spatial variability and significant divergences, which are attributed to several factors, including variations of the soil water content. The science behind this paper is of great interest to the development of remote sensing approaches to monitor permafrost in remote areas. Overall, I believe that this article could be published after some suggested modifications. The following provides my specific comments and suggests technical corrections to the manuscript.

*Specific comments:* My specific comments are organized with respect to the main parts of the manuscript.

**Comment:** *Introduction and previous work*: In my opinion, the introduction is light and could be improved. First, the beginning of the introduction (lines 9-14) does not highlight enough why remote sensing can be and important tool to monitor the evolution of permafrost. I'm not that convinced by that part and providing detailed explanations on why and how remote sensing tools can contribute to

the monitoring of ground-surface deformations and other permafrost related features is paramount. This can be in part achieved by moving most of the information provided in section 2 to the introduction. Section 2 provides an adequate review of remote sensing techniques mostly focussed to Alaska and permafrost terrains. Moving most of that information to the introduction would also help the authors highlight more the originality and contribution of their study to remote sensing and permafrost monitoring science – this should be clearly stated. Finally, the introduction as it is does not clearly state the main and specific objectives of the study. This should be done to help the reader understand the scope of the study.

**Answer:** This comment is addressed now in the revised version of the paper. We paraphrase part of the introduction to emphasize the importance of remote sensing (lines 20-23). Also, some information about ICESat-2 is now provided (lines 29-31). Moreover, the objectives of the study are clearly stated at the end of the introduction section (lines 31-35). The previous work section is shortened and the information is now in a table (Table 1).
* * *
**Comment:** *Study area*: The study area section could be detailed a little more. There are several site parameters that are not presented, which leaves several questions to the reader. Here are some specific topics that could be added to this section. (1) What are CALM and APEX sites? Are there physical equipment there? Monitoring stations? (2) Ground features focuses on vegetation, but what is the typical (range) of the stratigraphy? What is the typical depth of the water table? Topography? Thickness of the permafrost and range of the active layer in the region? These are important parameters. (3) In the same way, typical meteorological parameters describing the region would be appreciated. Air temperature, total and the partition of precipitations, radiation, thickness of the snow cover, etc. Again, these are important parameters describing a study site. (4) Why was this site selected?

**Answer:** Most of requested information is now added to study area section (lines 42-60), showing available data. We describe the CALM site with information about its topography, vegetation coverage, soil taxonomy, elevation of site, type of monitoring and organic matter thickness. These data are from metadata available from the CALM website, now referenced.
* * *
**Comment::** *Methodology, results and discussion*: Overall, the methodology associated to the InSAR processing workflow is well developed and adapted to the main goal of the paper. The displacements used to obtain the thickness of the active layer were obtained along the LOS, which could be in part a cause of the variability and disparities between the ATL and ADDT measurements. However, the authors nuance well why they believe this error is limited and I agree with this choice. In my opinion, one of the main downfalls of this approach is that a fairly complex approach was used to obtain the porosity models used to generate ALT inversions without demonstrating how important this parameter is (coupled with water content) to the results. In addition, the authors use air temperatures instead of ground-surface temperatures for their calculations – this could have had an important impact on the correlations. I understand that when no in situ data is available, one must used estimated values. This study is no different and the approach is ok. However, I would have appreciated some kind of parametric study highlighting at least the importance of the porosity model and water content and the resulting ALTs. I strongly believe such analyses (on selected sites) should be added to better understand the important parameters governing the approach that was developed. This being said, the authors made an effort to describe well these limitations. In addition, since the stratigraphy of the site(s) is not presented, it is hard to assess if the porosity model provided is realistic.

**Answer:** We agree with this comment, this is a limitation of any ALT model that relies on remotely sensed data. We address part of this comment in "limitations and future work" section in lines 351-361. The soil moisture in different layers of soil (shallow ground water) and its flow related to

microtopography are certainly important factors which are poorly understood. We can only assume that soil is fully saturated because the area is an ice-rich permafrost zone and is likely to remain saturated during the thaw season (our own sparse observations in the discontinuous permafrost zone of central Alaska, south of the study site, confirm saturated or near-saturated conditions near the end of the 2023 thaw season). High resolution microtopography, shallow hydrology, local rainfall, the porosity and bulk density of different soil horizons, organic matter characteristics, maximum root depth and other parameters are all potentially important but beyond the scope of the current study, which emphasizes what can be done with available satellite data and global models.
* * *
*Technical corrections*:

**Comment:** General technical consideration 1: This is more of a personal choice, but I would suggest avoiding the */we/* and *our/* wording. Use passive voice.

**Answer:** This is a style preference. We think in most cases active voice is simple, clear and shorter than re-arranging sentences to be in passive voice.
* * *
**Comment:** General technical consideration 2: A definition of what is the Alaskan North Slope could be added early in the manuscript.

**Answer:** This comment is addressed in the revised version of the paper (lines 42-45).
* * *
**Comment:** Line 1-2: I suggest specifying the study site / area.

**Answer:** This comment is addressed in the revised version of the paper.
* * *
**Comment:** Figure 1: add Latitude and Longitude axes to Figure. Specify elevation in m and LOS displacements in cm to legends in Figure 1.

**Answer:** This comment is addressed in the revised version of the paper (Figure-1).
* * *
**Comment:** Line 57: site is described as having 23 cm organic layer thickness : this is really specific and precise. Consider provide a range? What is the stratigraphy?

**Answer:** The organic layer thickness came from metadata of CALM site used in this study. We cited the website that has metadata in the manuscript. https://www2.gwu.edu/~calm/data/webforms/u8_f.htm

We assume that the whole area has the same organic layer thickness. Unfortunately, we could not find wide stratigraphic data which we believe it is necessary in terms of classifying porosity and bulk density in ALT models. We add this challenge in the limitation sections of the paper (See section 3 and 5.6).
* * *
**Comment:** Line 58: Seasonal high water table subject to saturation : not sure what this exactly means. … shallow water table subjected to seasonal variations? This is important.

**Answer:** This comment is now addressed in the revised version. The sentence is rephrased (lines 50-51).
* * *
**Comment:** Lines 61-63: This should be moved to the methodology section.

**Answer:** This comment is addressed in the revised version of the paper.
* * *
**Comment:** Line 68-69: Were there any actions taken to make sure no snow-covered acquisitions were used (i.e. visual validation based on photos)? What was the revisit period of those acquisitions? Which Track and sensing orbit?

**Answer:** This comment is addressed in the revised version of the paper. The information about revisit time of Sentinel-1 data (12 days) and its track (descending) were added to the manuscript (lines 67-70). We made sure that the data we used were several days after beginning of thaw season. In this case, for example in season 2017, the first Sentinel-1 SAR data used in our time series was acquired June 12, while thaw began here on May 24 according to the Sagwon meteorological site. The ADDT for this time period was ~48 Celsius day which we think was enough to confirm snow-free conditions. For the most part optical image data are limited by cloud cover.
* * *
**Comment:** Line 85: Figure S1: I believe this figure should be moved to the paper since it is referred a few times and is of good quality / indicative.

**Answer:** We think that since many of points are either located in decorrelated area of InSAR or have higher uncertainty/standard deviation, it is probably better to put it in the supplement materials.
* * *
**Comment:** Line 158-160: This should have been said in the study site section.

**Answer:** This comment is addressed in the revised version of the paper (lines 45-54).
* * *
**Comment:** Line 205: Maximum subsidence occurs at the end of the summer for all locations. It is to be said that monitoring was stopped (acquisitions were not considered) after the last point shown in Figure 3. This is attributed to freezing conditions catching up. Normally, we would see a gradual uplift from

then on, but this is not shown (for good reasons). I suggest explaining such phenomenon. Providing a comparison of air temperature and LOS displacement for one or selected stations could contribute to highlight such mechanism.

**Answer:** Part of this comment is addressed in the revised version of the paper. This is correct. The pattern of subsidence would differ after the end of the thaw season and we may see uplift due to freezing. However, the objective of this study was to only focus on different thaw seasons. Also, radar decorrelation happens with early precipitation (snowfall) at or near end of thaw season in this area.
* * *
**Comment:** Figure 5: Please use identical units for figure 1 and 5 (northing / easting / lat / long).

**Answer:** We are aware that different units in maps may confuse readers. But for showing the larger area like Figure-1 and its subset we preferred to show latitude and longitude, while small subset of area is shown in UTM coordinates. In processing step – ALT estimation model – we had to convert lat/long to UTM coordinates. The subset plot showed the area in Figure-1.
* * *
**Comment:** Lines 232-233: This is important. Add supporting ideas around that topic.

**Answer:** This comment is now addressed in the revised version of the paper. The time-series of ALT of U8 site from 1996 to 2022 as well as statistical analysis were applied to support high spatial and temporal variability of ALT statement. Figure 7 and statistical analysis in section 5.3 lines 299-310 supports our results.
* * *
**Comment:** Lines 287-283: A brief statement of the limitations should be added to the conclusions since a high correlation between ALT and ADDT was not achieved.

**Answer:** This comment is addressed in the revised version of the paper (section 6).